



# The mesoscale eddy field in the Lofoten Basin from high-resolution Lagrangian simulations

Johannes S. Dugstad[1], Pål Erik Isachsen[2,3], and Ilker Fer[1]

[1]Geophysical Institute, University of Bergen and Bjerknes Centre for Climate Research, Bergen, Norway
[2]Department of Geosciences, University of Oslo, Oslo, Norway
[3]Norwegian Meteorological Institute, Oslo, Norway

**Correspondence:** Johannes S. Dugstad (johannes.dugstad@uib.no)

**Abstract.** Warm Atlantic-origin waters are modified in the Lofoten Basin in the Nordic Seas on their way toward the Arctic. An energetic eddy field redistributes these waters in the basin. Retained for extended periods, the warm waters result in large surface heat losses to the atmosphere and an impact on fisheries and regional climate. Here, we describe the eddy field in the Lofoten Basin by analysing Lagrangian simulations forced by a high-resolution numerical model. We obtain trajectories of particles seeded at three levels: near the surface, at 200 m and 500 m depth, using 2D and 3D velocity fields. About 200,000 particle trajectories are analyzed from each level and each simulation. Using multivariate wavelet ridge analysis, we identify coherent cyclonic and anticyclonic vortices in the trajectories and describe their characteristics. We then compare the evolution of water properties inside cyclones and anticyclones as well as in the ambient flow outside vortices. As measured from Lagrangian particles, anticyclones have longer lifetimes than cyclones (16-24 days compared to 13-19 days), larger radius (20-22 km compared to 17-19 km) and a more circular shape (ellipse linearity of 0.45-0.50 compared to 0.51-0.57). The angular frequencies for cyclones and anticyclones have similar magnitudes (absolute values of about $0.05f$). The anticyclones are characterized by warm temperature anomalies whereas cyclones are colder than the background state. Along their path, water parcels in anticyclones cool at a rate of $0.02$–$0.04^{o}$C/day while those in cyclones warm at a rate of $0.01$–$0.02^{o}$C/day. Water parcels experience a net downward motion in anticyclones and upward motion in cyclones, often found to be related to changes in temperature and density. The along-path changes of temperature, density and depth are smaller for particles in the ambient flow. An analysis of the net temperature and vorticity fluxes into the Lofoten Basin shows that while vortices contribute significantly to the heat and vorticity budgets, they only cover a small fraction of the domain area (about 6%). The ambient flow, including filaments and other non-coherent variability undetected by the ridge analysis, hence plays a major role in closing the budgets of the basin.

## 1 Introduction

The Lofoten Basin (LB) in the Norwegian Sea is an important region for the retention and modification of the warm Atlantic Water (AW) as it flows northward towards the Arctic Ocean (Koszalka et al., 2011; Mauritzen, 1996; Rossby et al., 2009b). In this region, the AW splits into two branches (Poulain et al., 1996; Orvik and Niiler, 2002) (Figure 1 a), the Norwegian Atlantic Slope Current (Slope Current hereinafter) and the Norwegian Atlantic Front Current (Front Current hereinafter). These currents



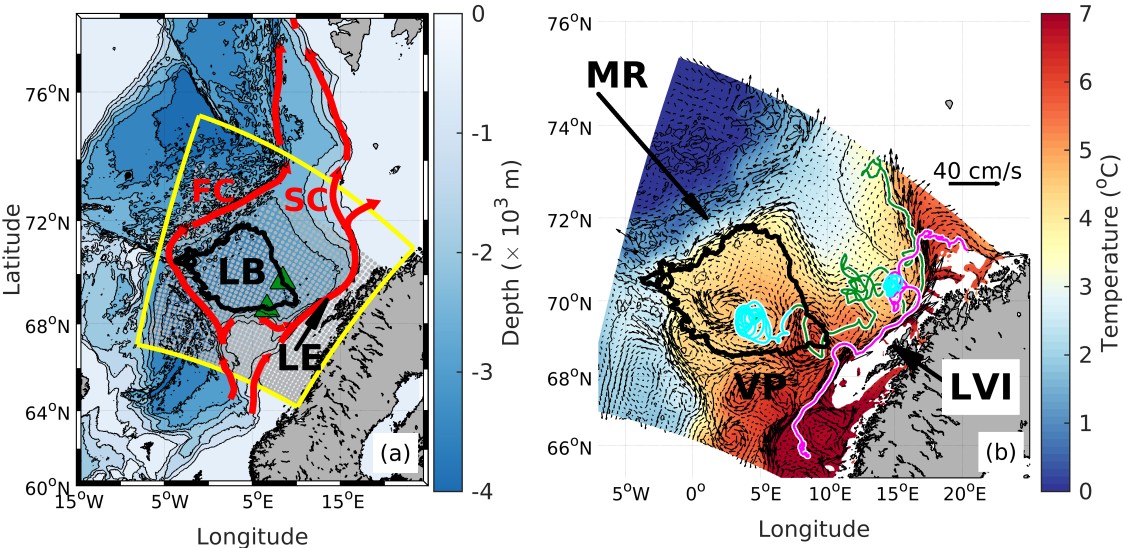

**Figure 1.** (a): Bathymetry of the Nordic Seas together with the main pathways of the Atlantic Water (red), the ROMS model domain (yellow) used for the OpenDrift simulations, and the deployment grid of the drifters (gray dots). Highlighted black contour shows the LB approximated by the 3000 m isobath with green triangles marking the edges of the southeastern part (see Section 4.3); (b): A zoom on the model domain showing the Eulerian mean temperature (colorbar) and the mean velocity (arrows, scale on the upper right) at 200 m depth, time averaged between 1996 and 2000. A couple of 2D-trajectories at 200 m are superimposed (green and magenta), with the ridge segments marked in cyan. Abbreviations are: LB=Lofoten Basin, LE=Lofoten Escarpment, FC=Front Current, SC=Slope Current, MR=Mohn Ridge, VP=Vøring Plateau, LVI=Lofoten-Vesterålen Islands

enclose the LB in the east along the continental slope off Norway and in the west along the Mohn Ridge, respectively. The warm AW spreads into the basin between these two branches. The LB is therefore a major heat reservoir in the Nordic Seas (Nilsen and Falck, 2006), and its heat content has increased over the last three decades (Broomé et al., 2020). Exposed to increased residence times, winter cooling and vertical mixing, the AW layer in the basin thickens and can reach depths of 500 m (Bosse et al., 2018; Mauritzen, 1996).

Both the Slope Current and the LB (here defined as the central basin enclosed by the 3000 m isobath, highlighted black contour in Figure 1) are characterized by energetic eddy fields. The Slope Current, in particular, is favorable for baroclinic instability as it passes very steep topography off the Lofoten-Vesterålen Islands (Köhl, 2007; Isachsen, 2015; Fer et al., 2020). This results in enhanced eddy-kinetic energy (EKE) levels (Koszalka et al., 2011; Andersson et al., 2011; Volkov et al., 2015; Fer et al., 2020) and large lateral diffusion rates (Koszalka et al., 2011). The EKE field in the LB is particularly strong towards

the center of the basin. There, at a mean position of around $[3^oE, 70^oN]$, the Lofoten Basin Eddy (LBE, also referred to as the Lofoten Vortex) (Ivanov and Korablev, 1995; Köhl, 2007; Fer et al., 2018; Raj et al., 2015) appears as a permanent anticyclone with a relative vorticity between $-0.7f$ (Fer et al., 2018) and $-f$ (Søiland and Rossby, 2013; Fer et al., 2018), where $f$ is





the Coriolis frequency. A secondary EKE maximum is observed towards the southeastern boundary of the 3000 m isobath surrounding the basin.

The vigurous eddy field is thought to have an impact on the thickening of the AW layer in the LB and towards the slope. For example, studies of regional hydrographic observations (Rossby et al., 2009b; Bosse et al., 2018) have shown that isopycnals that define the AW layer reach the deepest levels in the LBE and in the secondary EKE maximum. The deep reaching AW is particularly studied in the LBE and has been related to a vertical stacking of lighter anticyclones that interact with the LBE and push the AW in the LBE further down (Trodahl et al., 2020). In addition, the LBE as well as other energetic eddies are

hypothesized to enhance small-scale turbulence, thereby mixing AW to deeper levels and also causing irreversible changes to the AW properties (Volkov et al., 2015; Bosse et al., 2018). While regionally-integrated transformation of AW can be estimated from hydrographic datasets, property changes of water parcels *along* their trajectories, as well as the role of mesoscale eddies versus other flow features along the trajectories, remain largely unknown.

The exchange of AW with the LB (AW-LB exchange) has also been studied extensively. Earlier literature has suggested

that much of the warm AW in the LB stems from anticyclonic eddies in the form of long-lived coherent vortices that shed off from the Slope Current and then bring with them heat and vorticity westward into the basin (Köhl, 2007; Volkov et al., 2013; Isachsen et al., 2012; Raj et al., 2016). However, how far these vortices are advected and how often they reach deep into the LB is uncertain. Raj et al. (2016) used satellite altimetry to estimate that about 75% of cyclonic and anticyclonic vortices in the region have a lifetime shorter than 30 days and that their drift speed never exceeds 8.3 cm/s. To cover a typical distance

from the slope to the center of the basin within a lifetime of 30 days, an eddy translation speed of 16 cm/s is required, which exceeds the upper value found by Raj et al. (2016). This suggests that other processes may have to contribute significantly to the AW-LB exchange. Indeed, previous Lagrangian studies (Dugstad et al., 2019a, b; Koszalka et al., 2011) have shown that surface drifters carried in a broad slab of water in the southern sector of the basin interact with the LB. And an Eulerian analysis by Dugstad et al. (2019a) showed that a key surface inflow contribution to the LB heat budget is primarily related to

the mean-flow component.

To describe and quantify the role of the eddy field for the AW-LB exchanges it is important to note that energetic vortices not only carry around the properties trapped in their cores but also stir and transport more passive water masses surrounding them. In an idealized simulation of an unstable eastern boundary current over steep topography with a deeper basin to the west (mimicking the domain we study here), Spall (2010) identified that narrow structures or 'filaments' surrounding anticyclonic

eddies can carry large cyclonic vorticity and hence make an important contribution to the net vorticity flux to the basin. It seems plausible that transport of such filaments is important to the heat and vorticity budgets also of the real Lofoten Basin, but to our knowledge a quantification of this has not been done before.

In this work we study the eddy field around the LB from a Lagrangian perspective. We perform Lagrangian simulations forced by high-resolution model outputs and extract eddy signals from synthetic particle trajectories using the method of

multivariate wavelet ridge analysis (Lilly et al., 2011; Lilly and Olhede, 2009). We will thus distinguish between eddies and the ambient flow. By 'eddies' we will be referring to coherent vortices in which trapped particles undergo repeated orbits, or oscillations, within a range of time and space scales. The 'ambient flow' will then refer to all features other than the coherent




vortices, including large-scale mean flows, filaments and other smaller-scale non-coherent flow features. The study consists of three main parts: (1) a quantification of the eddies and their characteristics (shape, size, rotation speed, spatial distribution), (2) 75 a comparison of how the characteristics of water masses in eddies and the ambient flow evolve with time, and (3) calculations of the net temperature and vorticity fluxes into the LB with an assessment of the relative contribution from eddies and the ambient flow.

## 2   Data and Methods

### 2.1   Ocean Model

The Lagrangian trajectories are integrated by using outputs from a high-resolution Regional Ocean Modelling System (ROMS) configuration in the Nordic Seas. ROMS is a hydrostatic model solving the primitive equations on a staggered C-grid with terrain-following vertical coordinates (Shchepetkin and McWilliams, 2009; Haidvogel et al., 2008). We use a fourth-order centered scheme for vertical advection, and a third-order upwind scheme for horizontal tracer and momentum advection. No explicit horizontal eddy viscocity or diffusion is applied, but the upwind advection scheme exhibits implicit numerical 85 diffusion. Vertical mixing processes that are not resolved by the model grid are parameterized by the $k - \epsilon$ version of the General Length Scale scheme (Umlauf and Burchard, 2003; Warner et al., 2005). The open lateral boundaries are relaxed toward monthly fields from the Global Forecast Ocean Assimilation Model (MacLachlan et al., 2015) and the atmospheric forcing is provided by 6-hourly fields from the ERA-Interim atmospheric reanalysis (Dee et al., 2011). The model has an 800 m horizontal grid resolution and 60 vertical layers. The vertical resolution is 2 to 5 m near the surface and 60 to 70 m 90 towards the bottom. Model outputs are stored every 6 hours between January 1996 and January 2000. This spatial and temporal resolution allows the model to resolve the mesoscale, and to some extent sub-mesoscale, features (Isachsen, 2015; Trodahl and Isachsen, 2018).

### 2.2   Lagrangian simulations

The Lagrangian simulations are the same as in Dugstad et al. (2019b). We use OpenDrift (Dagestad et al., 2018), an open source 95 Python-based framework for Lagrangian modelling which operates offline by using a stored model velocity output. We perform two experiments, one using only the horizontal velocity (2D experiment), and a second one using the full three-dimensional velocity field (3D experiment). The Lagrangian positions (longitude, latitude, depth) are updated using the 6-hourly model currents by applying a fourth-order Runge-Kutta integration routine and are stored at 6 hour intervals. Potential temperature, salinity and velocity fields are linearly interpolated onto the particles. We also create daily fields of relative vorticity and 100 Okubo-Weiss fields (see Section 3.1) and interpolate these similarly. We do not add explicit lateral or vertical diffusion to the drifters. This is to avoid the trajectories to be too diffusive with respect to transport properties of the ROMS model used to force them (Dugstad et al., 2019b). The trajectories were tested and compared with 2D surface drifters and found to reproduce the relative dispersion of surface drifters very well (Dugstad et al., 2019b).





In all simulations we deploy particles at three levels (15 m, 200 m and 500 m) in sets of 1600 particles every week for three years, from 1 January 1996 to 1 January 1999 (deployment positions are shown in Figure 1 a). In total, this gives 156 weeks of deployments, and 1600×156=249,600 particles at each deployment depth. The particles are given a lifetime of 1 year, i.e., the trajectory data end on 1 January 2000. We remove all particles that are deployed in areas shallower than 200 m. After excluding these, the number of particles are reduced to 225,000 at 15 m and 200 m and 195,000 at 500 m for both the 2D and 3D simulation. For simplicity, we will refer to the Lagrangian particles as "drifters", and use "temperature" and "density" for

potential temperature and potential density.

### 2.3 Multivariate wavelet ridge analysis

To identify long-lived coherent vortices in the Lagrangian trajectories, we perform a *multivariate wavelet ridge analysis* (MWRA), which is developed for the purpose of finding "loops" in drifter trajectories and hence identify whether the drifters are inside coherent vortices or not. We describe the basic concepts of the method here, but more details can be found in Lilly

and Olhede (2009); Lilly et al. (2011).

A drifter deployed in position $(x_o, y_o)$ that moves around with time will experience east/west, $x(t)$ and north/south, $y(t)$ displacements relative to the deployment position. The vector valued function $\mathbf{z}(t) = \begin{bmatrix} x(t) \\ y(t) \end{bmatrix}$ thus represents the displacement signal for the drifter, relative to the deployment position. Loops in a drifter trajectory would occur as an oscillating signal in the time series $\mathbf{z}(t)$. The routine seeks to trace this oscillation by performing a wavelet transformation to the time series $\mathbf{z}(t)$:

$$\mathbf{w}_{\mathbf{z},\psi}(t,s) = \int_{-\infty}^{\infty} \frac{1}{s} \psi^* \left( \frac{\tau - t}{s} \right) \mathbf{z}(\tau) d\tau \tag{1}$$

where $\mathbf{w}_{\mathbf{z},\psi}$ is the wavelet transform, $\psi$ is the wavelet used with the asterisk denoting the complex conjugate, $s$ is a scaling factor that controls the contraction or dilation of the wavelet in time, $\tau$ is the time which we integrate over, and $t$ is the shift as the wavelet is shifted in time along $\mathbf{z}$. The wavelet transform $\mathbf{w}_{\mathbf{z},\psi}$ can be regarded as a projection of the time series $\mathbf{z}$ onto the wavelet $\psi$ for different choices of $t$ and $s$. Hence, large values of $\mathbf{w}_{\mathbf{z},\psi}$ are expected where the projection of $\mathbf{z}$ onto $\psi$ is good,

that is in regions where $\mathbf{z}$ experiences oscillations similar to the oscillations given by $\psi$ for a given $t$ and $s$. In other words, given the right conditions on $t$ and $s$, when a looping drifter trajectory leads to oscillations in $\mathbf{z}$, this will result in large values of $\mathbf{w}_{\mathbf{z},\psi}$.

The MWRA routine looks for "ridge points". A ridge point is defined as a point on the time/scale plane that satisfies

$$\frac{\partial}{\partial s} ||\mathbf{w}_{\mathbf{z},\psi}(t,s)|| = 0, \ \frac{\partial^2}{\partial s^2} ||\mathbf{w}_{\mathbf{z},\psi}(t,s)|| < 0 \tag{2}$$

meaning ridge points are locations where the norm of the wavelet transform vector experiences a local maximum with respect to the scale $s$. Persistent oscillations in $\mathbf{z}$ with time will thereby lead to a continuous curve of adjacent ridge points in time that are connected to each other. We refer to this curve as a "ridge". Per definition, the ridge traces out the signals with the largest intensity/energy in the wavelet transform. Therefore, when a drifter contains a ridge, this is interpreted as that the drifter is caught inside an eddy, given our parameter choices (see below). For more details, see Lilly et al. (2011).



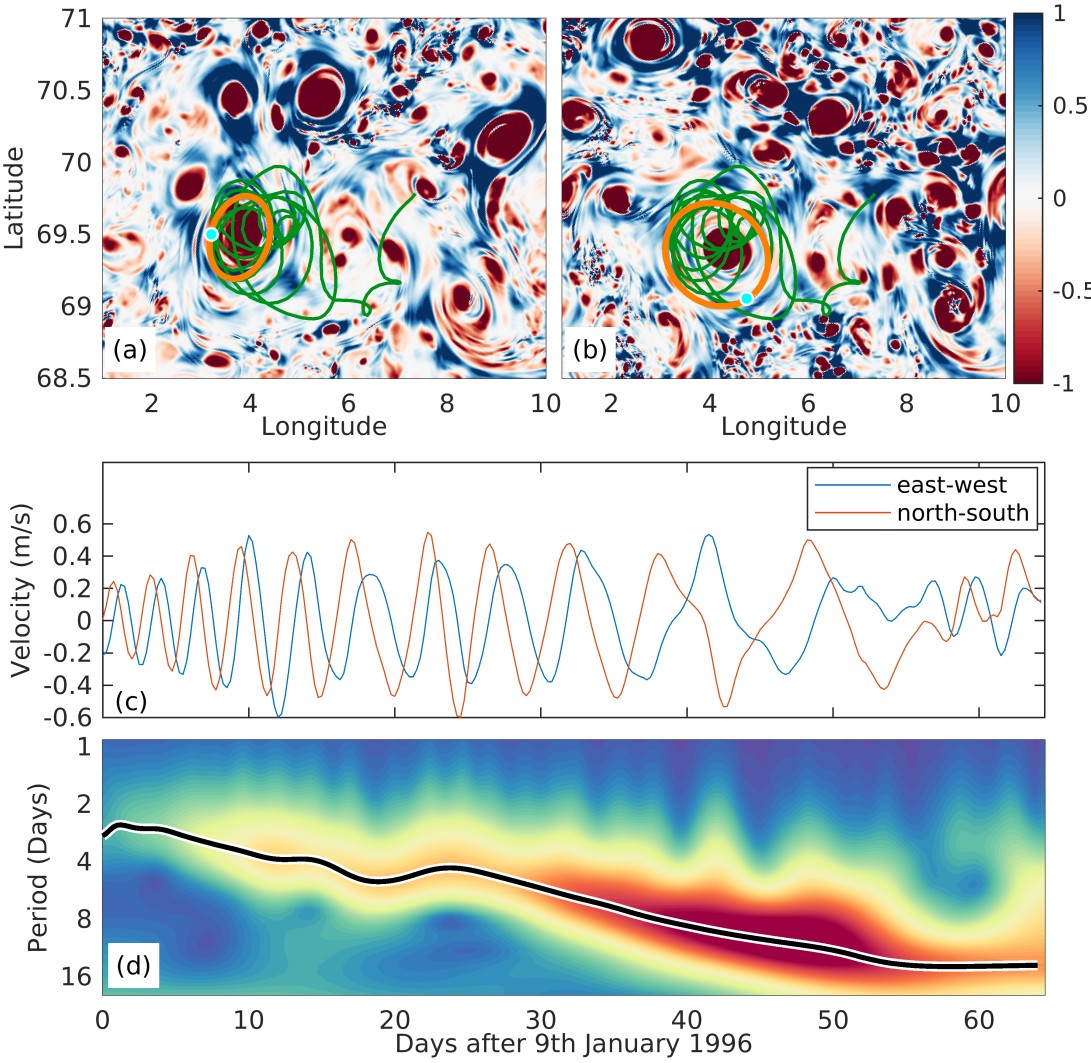

**Figure 2.** (a,b): The basin trajectory (the ridge segment) from Figure 1 b superimposed on snapshots of the OW field (see text) from (a) 1 February 1996 and (b) 15 February 1996. Cyan dots and orange ellipses show the position of the drifter and the instantaneous spanned out ellipse (see text) as computed from the MWRA routine at the two dates. Colorbar unit is $10^{-9}$ s$^{-2}$. Example of the MWRA routine applied to the trajectory is shown in c,d. (c): The east-west and north-south velocities ($u$ and $v$) of the trajectory; (d): The magnitude of the total wavelet transform $||\mathbf{w}_{\mathbf{z},\psi}(t,s)|| = \sqrt{|w_+(t,s)|^2 + |w_-(t,s)|^2}$ where $|w_+(t,s)|$ and $|w_-(t,s)|$ are the absolute values of the positive and negative rotary transforms respectively. The black line indicates the ridge which is given by the maximum of $||\mathbf{w}_{\mathbf{z},\psi}(t,s)||$.

After identifying the ridges, the routine gives the output of the longitude and latitude of each ridge. Residual longitude and latitude, that is longitude and latitude along the ridge after subtracting the eddy signal, are computed to represent the location of the mass center of the eddy tracked by the drifter. At a given position and time along a ridge, the curve traced out by its





**Table 1.** List of output from the MWRA routine. In descriptions, $a$ and $b$ are the semi-major axis and the semi-minor axis of an ellipse.

| Variable | Description | Unit |
|---|---|---|
| **Signal fields** | | |
| lon | Longitude along each ridge | (°) |
| lat | Latitude along each ridge | (°) |
| lonres | Longitude residual along each ridge after subtracting eddy signal | (°) |
| latres | Latitude residual along each ridge after subtracting eddy signal | (°) |
| **Ellipse fields** | | |
| $\lambda$ | Ellipse linearity, $\lambda = \frac{a^2 - b^2}{a^2 + b^2}$ | |
| $\omega$ | Ellipse instantaneous angular frequency | (radians/day) |
| $R$ | Ellipse geometric mean radius | (km) |
| $V$ | Ellipse orbital velocity | (cm/s) |
| $\omega_n$ | Ellipse non-dimensional frequency, $\omega_n = \frac{\omega}{f}$ | |
| RL (loops) | Ridge length in terms of loops | |
| RL (days) | Ridge length in terms of days | (days) |

instantaneous motion can be described as an ellipse with major axis $a$ and minor axis $b$. We obtain such ellipse parameters at each instantaneous position on a ridge (every ridge point) to identify the typical characteristics of the eddies (size, shape, rotation speed etc.) in the region. A summary of the drifter variables used in this study is given in Table 1.

An example of the MWRA routine applied to the ridge (cyan) of the green trajectory in Figure 1 is given in Figure 2. This drifter was deployed close to the center of the LBE on 8 January 1996, and immediately started to loop around, leading to the detection of the ridge from 9 January to 14 March 1996. The ridge plotted over snapshots of the Okubo-Weiss field (described in Section 3.1) shows that the drifter loops around the LBE (shown by negative/red values/colors). The east-west (u) and north-south (v) velocities (Figure 2 c) show larger periods with time, suggesting that the drifter loops with larger radius as time goes. The magnitude of the total wavelet transform $||\mathbf{w}_{\mathbf{z},\psi}(t,s)||$ (Figure 2 d) shows a persistent region of high intensity indicating adjacent ridge points. Thus, the maximum of $||\mathbf{w}_{\mathbf{z},\psi}(t,s)||$ traces out a well defined ridge (black curve), indicating an oscillation period increasing with time. The ellipse parameters (Table 1) are then obtained by analyzing the instantaneous motion of the drifter along the ridge to estimate the size, shape, rotation speed etc. of the eddy. Two exemplary ellipses are plotted for 1 February 1996 and 15 February 1996 (Orange ellipses in Figure 2 a and b respectively). The size of the ellipses increase with time, again indicating that the drifter loops with larger radius with time.

For each drifter the routine may find zero, one or several ridges that are given with indices along the drifter trajectories. This means that for a given drifter trajectory, we are able to identify where and when the drifter experiences ridges. By applying the routine to a large number of drifter trajectories, we can statistically identify patterns of the eddy activity. Furthermore,




since we track the drifters before and after they experienced ridges, we can interpret how eddies affect the path and water mass
properties along the drifter trajectories.

   To objectively select ridges that are associated with eddies, some choices are made. Before running the MWRA routine, we
choose a frequency band of $1/64 < |\omega_n| < 1$ (i.e., we only allow oscillations in $\mathbf{z}(\tau)$ within this frequency band), similar to
Lilly et al. (2011), where the non-dimensional frequency $\omega_n = \omega/f$ is the ratio of the angular frequency $\omega$ of the vortex/eddy

as sampled by drifters and the local Coriolis frequency $f$. The lower limit allows to capture oscillations far out on an eddy
flank while the upper limit is sufficient to capture the nonlinear mesoscale features in the region. Note that $|\omega_n|$, in solid body
rotation equals to half of the Rossby number $|\frac{\zeta}{f}|$, where $\zeta$ is the vertical component of relative vorticity. Nonlinear features
such as the LBE having $|\frac{\zeta}{f}|$ =0.7-1, would therefore be captured by this band. In fact, $|\omega_n|$ never exceeds 0.52, meaning that
choosing a frequency band of $1/64 < |\omega_n| < 0.52$ would not change our results. An important choice is the minimum ridge

length threshold. We assign the ridge length both in terms of time (how long the drifters experience a ridge), and in terms of
the number of cycles the drifters experience along the ridge.

   This second threshold is chosen as a function of the number of oscillations within the wavelet we use, leading to the problem
of first choosing wavelet duration $P_{\beta\gamma} = \sqrt{\beta\gamma}$ where $\beta$ and $\gamma$ are constants controlling the form of the wavelet (Lilly and
Olhede, 2012). The wavelet duration is defined such that $2\frac{P_{\beta\gamma}}{\pi}$ is approximately the number of oscillations contained within

the wavelet in the time domain (Lilly and Olhede, 2009) and the number of oscillations is controlled by the parameters $\beta$ and $\gamma$.
We follow Lilly and Olhede (2012) and Lilly et al. (2011) who state that a good choice would be the so-called "Airy" wavelet
giving a wavelet duration of $P_{\beta\gamma} = \sqrt{2 \cdot 3} = \sqrt{6}$. This value gives a high degree of time concentration at some expense of the
frequency resolution, but captures the features we are looking for. To accurately identify eddies in our ridges, we set the ridge
length (in terms of cycles) to $2\frac{2P_{\beta\gamma}}{\pi} \approx 3.1$. This is a rather strict threshold and twice as large as the value set in Lilly et al.

(2011). However, they find that ridges with fewer or the same amount of cycles as their set threshold (1.6) are usually spurious
because one looks for ridges with fewer cycles than the wavelet itself. The ridges get statistically more significant by increasing
the threshold.

   In terms of time, we choose a minimum ridge length of 2 days. However, due to the choice of ridge length in terms of cycles,
this latter choice did not affect the results. Lastly, we also remove about $\frac{P_{\beta\gamma}}{\pi}$ oscillations from each end of a ridge to exclude

the spin-up effects at the edges (Lilly et al., 2011). As a result, the ridges can have a minimum of 1.6 cycles. We further discuss
the ridge length in Sections 3.1 and 4.1.

   After running the routine, the orbital velocity ($V$) and geometric radius ($R$) from every ridge point are bin averaged on the
radius/velocity ($R/V$) plane (Figure 3). We show the result for 3D drifters deployed at 200 m, but other depths are qualitatively
similar. Orbital velocities increase with radius until a maximum of about 60 cm/s at radii of about 25 km. Ridges with a larger

radius are therefore often found on eddy flanks. The anticyclones have a slightly larger maximum orbital velocity, and a
smaller ellipse linearity ($\lambda$) than cyclones implying stronger and more circular anticyclones. We observe high ellipse linearity
at very small radii. Since we are interested in mesoscale features, we discard all ridge points with $\lambda > 0.95$ and $R < 5$ km. This
removes some small scale loops in the trajectories, especially at 500 m depth, which we verified by inspection that are caused
by artifacts. Note that the latter condition also will remove parts of ridges that normally loop with $R > 5$ km, but at some point





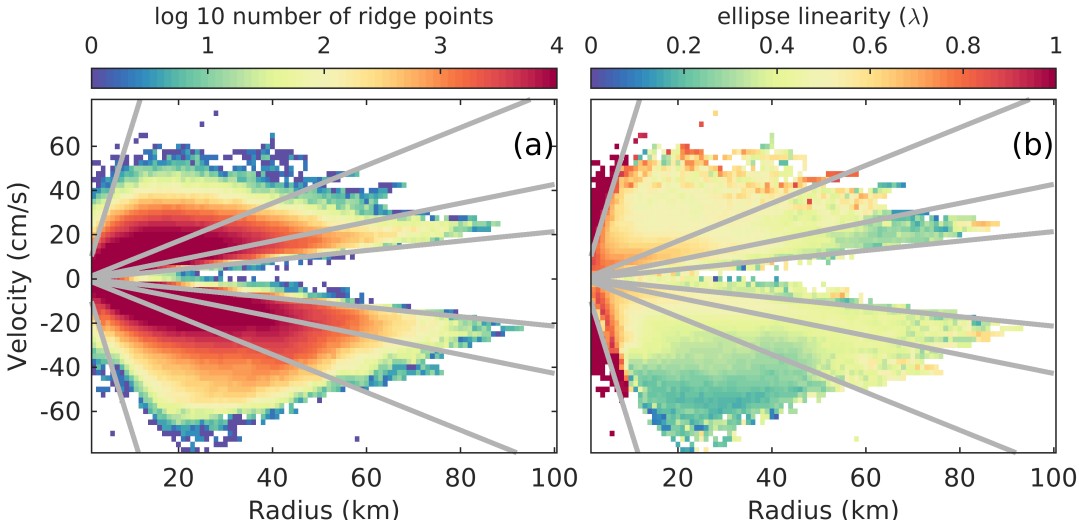

**Figure 3.** Distribution of ridge points in the radius/orbital velocity ($R/V$) plane for 3D drifters deployed at 200 m. Positive and negative velocities correspond to cyclonic and anticyclonic motions, respectively. Colors mark (a) $\log_{10}$ of the number of ridge points and (b) binned ellipse linearity onto the $R/V$ plane. Gray lines in both panels indicate different angular frequencies and are given as $\omega = \pm\frac{1}{64}f$, $\omega = \pm\frac{1}{32}f$, $\omega = \pm\frac{1}{16}f$ and $\omega = \pm\frac{1}{2}f$. Note that the slope $V/R$ is equivalent to $\omega$.

move closer to the eddy core ($R <$5 km). However, setting this condition only reduced the number of ridge points with about 5% at 15 and 200 m and about 10% at 500 m. Since we consider the majority of these data points as bad, we continue to use the condition. We discuss this in Section 4.2.

We compare three groups: Cyclonic and anticyclonic ridges, and the ambient flow. In the following, it is implied that a "ridge" is similar to a vortex and we will refer to these as cyclonic ($\omega_n >$0) and anticyclonic ($\omega_n <$0) ridges/eddies. The drifter

data points without ridges are then used to describe the ambient flow and will be referred to as AM drifters. Note that the AM class includes segments of any drifter trajectory which are not ridges.

## 3   Results

### 3.1   Vortex detection

Coherent vortices, detected as ridges, cover a small fraction of the total drifter data points. The evolution of fraction of ridge

points for each day after deployment, normalized by the number of available drifter data points for the given days, shows that about 6% of all drifter data points are ridges (Figure 4 c). This implies that a larger fraction (about 94%) of drifter data points are not vortices. Since a large number of drifters are distributed in the domain, we consider the fraction of ridge points as a proxy for the areal fraction of the domain that is covered by eddies. To study whether this fraction is representative, we compare this



with results obtained after analysing the Okubo-Weiss parameter OW$=\left(\frac{\partial u}{\partial x}+\frac{\partial v}{\partial y}\right)^2+\left(\frac{\partial v}{\partial x}+\frac{\partial u}{\partial y}\right)^2-\left(\frac{\partial v}{\partial x}-\frac{\partial u}{\partial y}\right)^2$ (see Penven

et al. (2005); Raj et al. (2016); Trodahl and Isachsen (2018) for a further explanation of the OW parameter). This quantifies the

relative importance of the strain, shear and relative vorticity of the flow. Grid cells with OW$< 0$ indicate cells where the relative

vorticity dominates and can thus be representative of vortex cores. Note however that OW$< 0$ is not a sufficient criterion to

conclude whether the cell is inside an eddy or not. A more stringent requirement which is commonly used is OW$< 0$ inside

closed sea surface height (SSH) contours (Trodahl and Isachsen, 2018; Raj et al., 2016).

We obtain a mean OW field from the ROMS model using daily fields of OW between 1 January 1996 and 1 January 2000,

and then averaging these in time. The same daily fields of OW are interpolated to the drifter trajectories. Maps shown from

the ROMS model and the 2D drifters (after binning) at 15 m indicate different results (Figure 4 a,b). In the ROMS model, the

LBE, as well as other eddy-like features towards the slope are resolved. However, the drifters tend to mainly trace positive OW

values. We compute the fraction of grid points with OW$<0$ from the ROMS model (solid green line in Figure 4 c) for each day

during year 1999 (other years are similar) and compare this to the fraction of drifter data points that sample OW$<0$ for each day

after deployment (dashed green lines). The fraction of drifter data points with OW$<0$ are given after requiring that it should

be less than 0 for at least 0, 1 or 2 consecutive days (Figure 4 c). In addition, we split any occurrence of OW$<0$ along drifter

trajectories into cyclonic (positive relative vorticity) or anticyclonic (negative relative vorticity). The ROMS model resolves a

higher fraction of OW$<0$ than the drifters. However, at the deployment time of the drifters the fractions are similar, implying

that the drifters (which are uniformly deployed) resolve the same OW field as the model at deployment.

Figure 4 will be discussed further in Section 4.1. Here we note that the fraction of OW$<0$ is 6 to 7 times larger in the ROMS

model than the fraction of ridge points found from the MWRA routine (compare solid green and black lines) which is about

6%. The different spatial OW distributions and the different fractions with negative OW of the ROMS model and the drifters,

could imply that the drifters are not realistically entrained into the eddies. The drifters therefore might undersample the eddies.

However, it is important to keep in mind that OW$<0$ is not a sufficient criterion to identify eddies, and the fraction of eddies

from ROMS is an overestimate, hence likely closer to the estimates from the MWRA routine.

### 3.2 Vortex characteristics

We now focus on the coherent vortices found by the MWRA routine. Although the fraction of ridge points is relatively small,

about 30% of the drifters experiences ridges during their lifetime for all depths and simulations (see NOD in Table 2).

The number of ridges (NOR) after dividing into cyclonic and anticyclonic (Table 2) indicates more anticyclonic ridges than

cyclonic ridges for all depths in both 2D and 3D simulations, with the exception for 2D drifters at 15 m which have similar

fraction of cyclonic and anticyclonic ridges. The vertical motion induced by strong atmospheric cooling, convection and mixing

in winter, observed e.g., in the 3D trajectories in Dugstad et al. (2019b), are not captured by the 2D drifters, and eddies are

therefore sampled differently for 2D and 3D drifters deployed at 15 m. Deeper at 200 and 500 m the effect of surface cooling

is weaker and the two simulations give similar results. We show results from 3D drifters in this section.

Probability density functions (PDFs) of selected parameters are summarized in Figure 5 for cyclonic and anticyclonic ridges

separately. The PDFs for $\omega_n$ indicate that vortices with different polarization rotate with approximately the same angular




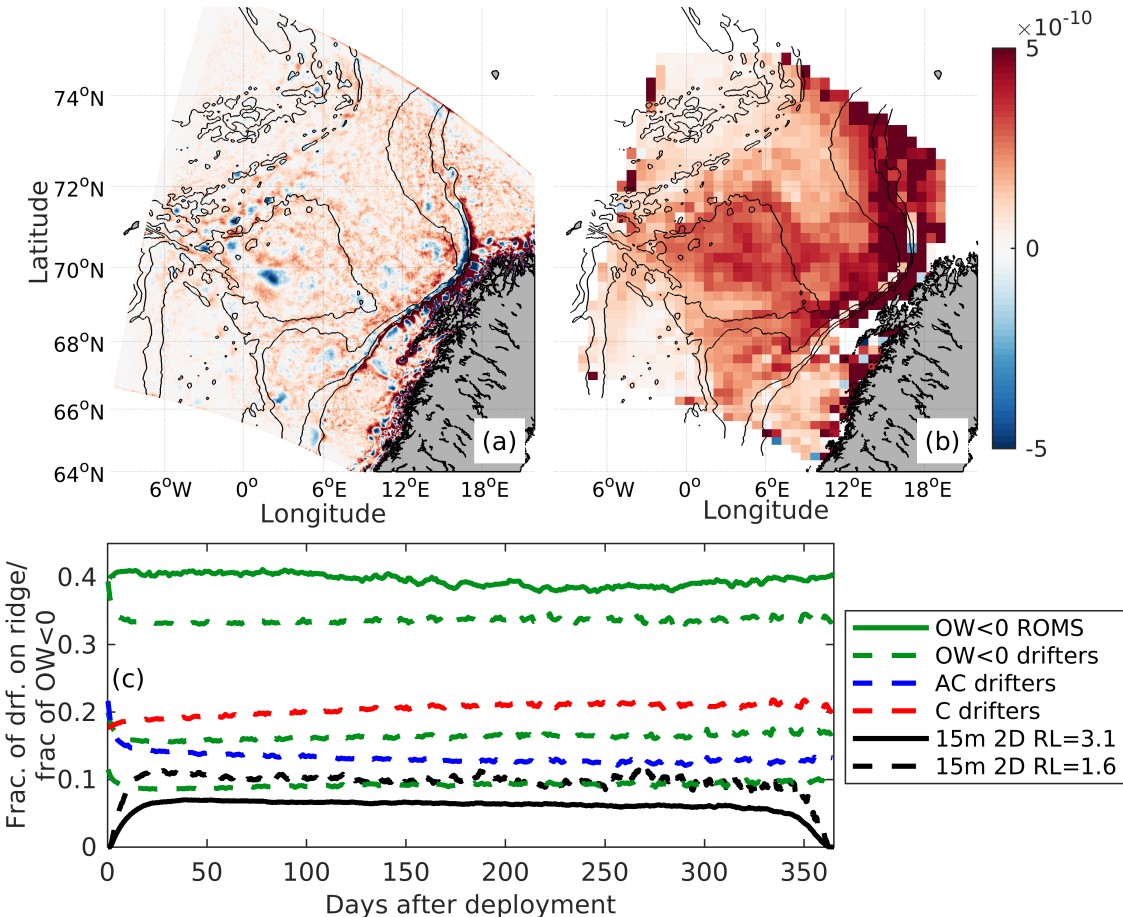

**Figure 4.** (a,b) The OW parameter (in s$^{-2}$) (a) from the ROMS model after averaging between 1996 and 2000 (same period as drifters) and (b) from the 2D drifters bin averaged after interpolating the ROMS fields to the drifters. (c) Evolution of the fraction of ridge points normalized to all drifter data points for 2D drifters deployed at 15 m, after running the MWRA routine with a minimum allowed ridge length (RL) of 3.2 cycles (solid black) and 1.6 cycles (dashed black). This is compared to the fraction of grid points with OW<0 from the ROMS model (solid green) at 15 m depth during a random year (1999) within the drifter simulation period. Dashed green lines show the fraction of data points along drifter trajectories with OW<0 normalized to the total number of drifter data points for every time step with the constraint that OW<0 for at least 0 days (upper), 1 day (middle) and 2 days (lower). Red and blue curves are for OW<0 with positive (red) or negative (blue) relative vorticity.

frequency, which is slightly smaller for drifters deployed at 500 m (Figure 5 a). Compared to typical Rossby numbers $\zeta/f$ of the LB region (see for instance Fer et al. (2018) and Søiland and Rossby (2013) showing $\zeta/f$ for the LBE), the values of $\omega_n$ are fairly small. However, note that $\omega_n = \omega/f = (V/R)/f$ (see Table 1 for explanations of the variables) while the Rossby number in cylindrical coordinates is $\zeta(R)/f = (V/R + \partial V/\partial R)/f$. The second term of this expression is not included in $\omega_n$ and the quantities are therefore not directly comparable. We discuss this further below in this section. Using the mean geometric radius





**Table 2.** Statistics for drifters in 2D and 3D simulations containing ridges, showing the number of drifters (NOD) that contained ridges with corresponding percentages in parentheses of total drifters studied for the given deployment depth (DD; 15 m=225,732, 200 m=224,016, 500 m=195,624). As a drifter can experience several ridges, the number of ridges (NOR) is also given, separately for cyclonic and anticyclonic ridges together with their percentage in parentheses. The non-dimensional frequency ($\omega_n$), mean geometric radius ($R$) and ellipse linearity ($\lambda$) after averaging over all ridge points for the given deployment depth and simulation (2D or 3D) are also given. The ridge length (RL) is listed both as an average of the number of loops exhibited by the drifter containing the ridge, and as an average of the number of days the drifters contain ridges.

| DD (m) | NOD (%) | NOR (%) | | $\omega_n$ | | $R$ (km) | | $\lambda$ | | RL (Loops) | | RL (Days) | |
|---|---|---|---|---|---|---|---|---|---|---|---|---|---|
| **2D** | | | | | | | | | | | | | |
| | | C | AC | C | AC | C | AC | C | AC | C | AC | C | AC |
| 15 | 63,691 (28) | 59,055 (49) | 62,180 (51) | 0.06 | -0.05 | 17.7 | 21.4 | 0.57 | 0.50 | 3.2 | 3.4 | 13 | 16 |
| 200 | 71,959 (32) | 55,278 (42) | 77,856 (58) | 0.05 | -0.05 | 18.7 | 21.9 | 0.56 | 0.48 | 3.1 | 3.7 | 16 | 20 |
| 500 | 57,663 (29) | 42,568 (43) | 56,080 (57) | 0.04 | -0.05 | 17.0 | 21.3 | 0.51 | 0.45 | 3.3 | 3.6 | 19 | 24 |
| **3D** | | | | | | | | | | | | | |
| 15 | 68,217 (30) | 55,998 (43) | 74,256 (57) | 0.05 | -0.05 | 18.1 | 21.8 | 0.57 | 0.48 | 3.1 | 3.6 | 14 | 18 |
| 200 | 73,874 (33) | 56,875 (41) | 80,774 (59) | 0.05 | -0.05 | 18.0 | 21.8 | 0.55 | 0.47 | 3.2 | 3.7 | 15 | 21 |
| 500 | 58,982 (30) | 48,201 (47) | 55,363 (53) | 0.05 | -0.04 | 16.7 | 20.8 | 0.51 | 0.46 | 3.3 | 3.6 | 18 | 24 |

($R$) as a proxy for the actual size of the associated eddies, we find that anticyclones are larger than cyclones. The ridge lengths are longer (both counted in days or in cycles) for the anticyclones (Figure 5 c,d), implying that anticyclones have a longer lifetime. The same result is obtained for the average values (Table 2). In addition, the ellipse linearity ($\lambda$) is smaller (indicating a larger amount of nonlinear vortices) for the anticyclones. Cyclones are relatively elongated while anticyclones are circular and have larger radius at all depths.

The shape, size and lifetime of eddies depend on which regions they are observed. Geographical distribution of ridges is estimated by obtaining density maps by counting the occurrences of ridge points in geographical bins of size $0.73^o \times 0.25^o$ (longitude bins are scaled with a factor $1/\cos(70^o)$ similar to Dugstad et al. (2019b)). Density maps (Figure 6) for cyclonic and anticyclonic ridges are consistent with EKE maps from satellite altimetry (Volkov et al., 2013; Fer et al., 2020) and from observed drifter data (Koszalka et al., 2011), all showing the anticyclonic structure of the Lofoten Basin Eddy (LBE) and an anticyclonic secondary EKE maximum at the southeastern corner of the LB. Some anticyclonic ridges are found off the slope at all depths, particularly at 200 m. However, over the slope between the 1000 and 2000 m isobaths, cyclones dominate. As also noted by Ivanov and Korablev (1995) and Köhl (2007), there are cyclones around the anticyclonic LBE.

The location of the first occurrence of a ridge in a trajectory is counted in the same geographical bins (thick black contours in Figure 6). Anticyclonic ridges are often first found in the center of the basin where the LBE is located, close to the anticyclonic structure of the southeastern corner of the LB, and sometimes off the slope, in agreement with earlier literature (Köhl, 2007;




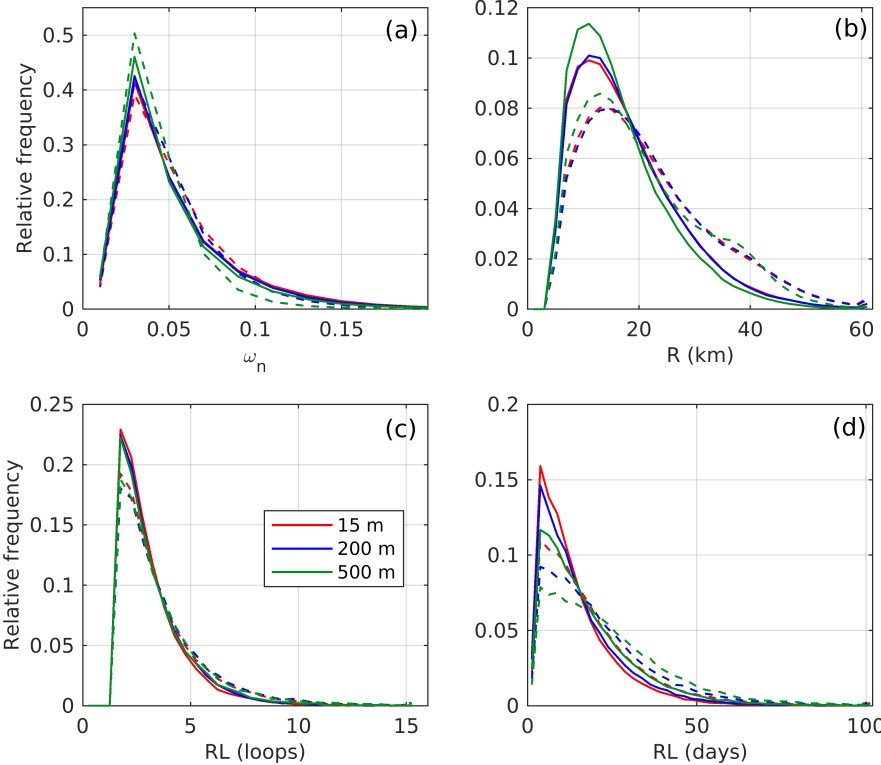

**Figure 5.** Probability density functions (PDFs) of (a) the non-dimensional frequency ($\omega_n$), (b) the mean geometric radius ($R$), i.e., the estimated instantaneous radius of the associated eddies, (c) length of the ridges in cycles, RL (loops) and (d) length of the ridges in days, RL (days). PDFs are for all ridges found from the 3D drifters deployed at 15 m (red), 200 m (blue) and 500 m (green) and shown for both cyclonic ridges (solid) and anticyclonic ridges (dashed). The PDFs are created after binning each quantity and normalized by the total number of cyclonic and anticyclonic ridge points, respectively.

Isachsen, 2015; Koszalka et al., 2011). Over the slope (around the 1000 m isobath) cyclones appear to be generated more fre-
quently. Note that the first ridge points may not always be representative of the eddy generation location, as in some occasions, the drifters could be deployed inside eddies. While this is likely the case in the permanent LBE, large number of first ridge points near the slope may give a realistic distribution of generation sites.

The spatial distributions of $R$, $V$ and $\lambda$ for anticyclonic and cyclonic ridges (Figure 7) are presented for 3D drifters deployed at 200 m (other depths are similar). They are obtained by averaging the quantities for all ridge points in longitude/latitude
bins. The largest eddies are anticyclonic and are found in the center of the LB associated with the LBE and with a mean geometric radius of 35-40 km, larger than the radius observed by Fer et al. (2018) (22 km) and Søiland and Rossby (2013) (18 km). However, they reported the radius of the maximum orbital velocities, hence drifters can loop around the LBE with a larger radius (i.e., on the flanks). The LBE is also characterized with large anticyclonic orbital velocity (about 30 cm/s) and





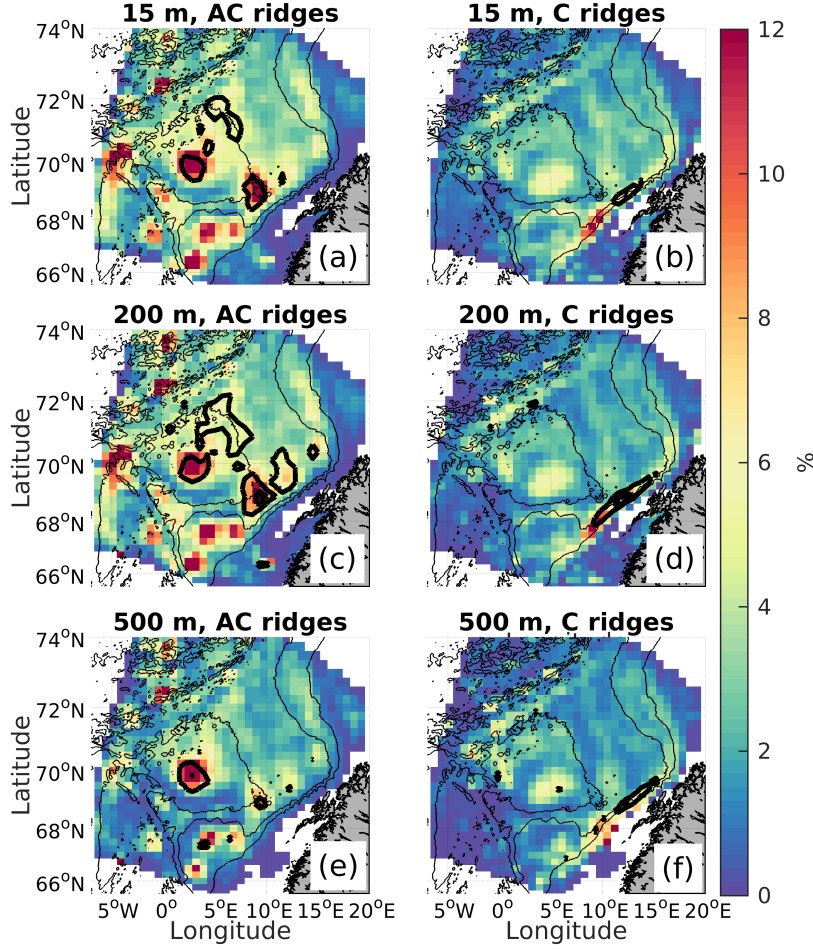

**Figure 6.** Density distribution of (a,c,e) anticyclonic ($\omega_n <0$) and (b,d,f) cyclonic ($\omega_n >0$) ridges for 3D drifters deployed at 15 m (a,b), 200 m (c,d) and 500 m (e,f). The colorbar shows the percentage of ridge points in each bin normalized to the total number of drifter data points in the same bins. Thin black contours show the 1000, 2000 and 3000 m isobaths, while thick black contours indicate key locations where the drifters are trapped into eddies, i.e., the first ridge point on each ridge. These contours are shown for more than 200, 300 and 400 first ridge points. A bin never contained more than 490 first ridge points. Bin sizes are $0.73^o \times 0.25^o$.

small ellipse linearity (about 0.3) (Figure 7 c,e). Note that due to large values of $R$, $\omega_n = (V/R)/f$ is enhanced relatively
less in the LBE (about 0.07-0.08, not shown). We observe that the values of $\omega_n$ in the LBE have similar magnitudes as the
Rossby numbers computed about 35 km from the eddy core of the LBE from observations (Figure 4 a in Fer et al. (2018)). As
discussed earlier in this section, $\omega_n$ is not directly comparable with the Rossby number, but similar magnitudes suggest that
the drifters trace reasonable values at the given radius. The relatively small magnitudes of $\omega_n$ in Figure 5 a (see discussion
above) therefore might be related to that the drifters often loop on the eddy flanks. We also find anticyclonic ridges towards the
southeastern corner of the LB and off the slope that have fairly similar properties to the LBE, i.e., increased orbital velocity




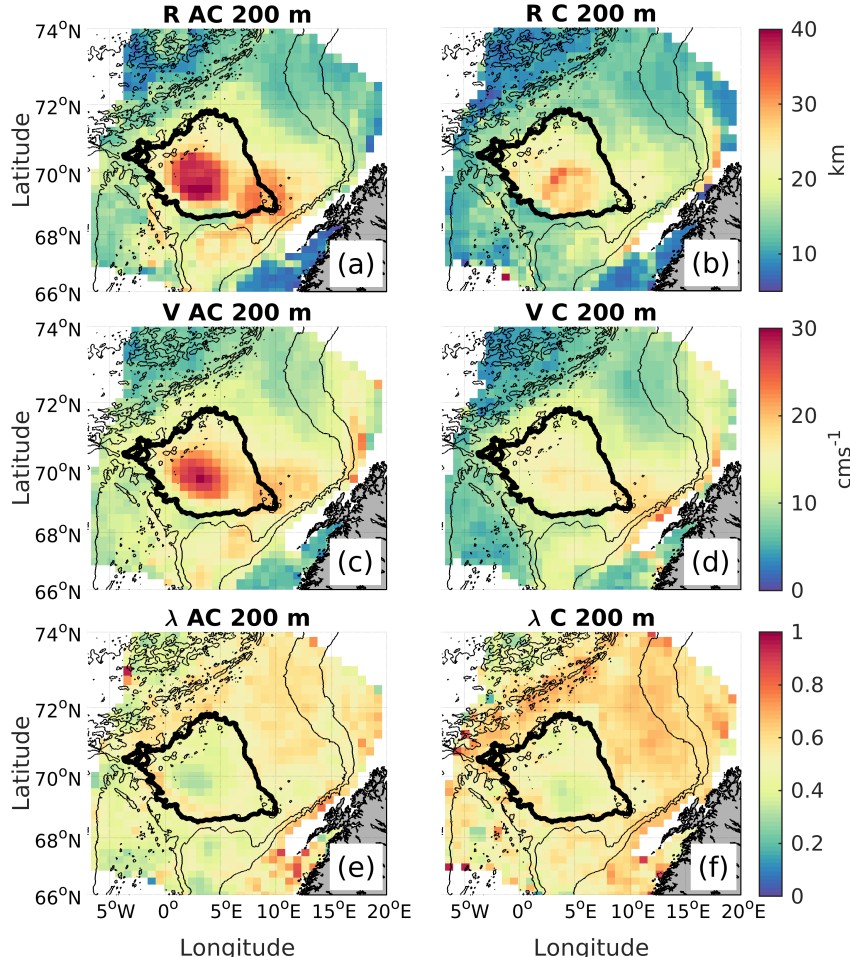

**Figure 7.** Averaged (a,b) mean geometric radius (km), (c,d) orbital velocity (cm s$^{-1}$) and (e,f) ellipse linearity for (a,c,e) anticyclonic and (b,d,f) cyclonic ridges from 3D drifters deployed at 200 m. Bin sizes are as in Figure 6. Thin black contours show the 1000 and 2000 m isobaths and the thick black contour indicates the LB approximated by the 3000 m isobath. Note that $V$ in panel c is negative, but plotted with opposite sign for better comparison.

and mean geometric radius, and decreased ellipse linearity, indicating a stable character. Over the slope (1000 and 2000 m isobath) off the Lofoten Escarpment, the cyclonic ridges show smaller radii and more elongated shape (higher $\lambda$), resulting in a more unstable character, possibly explaining why cyclones have shorter lifetime than anticyclones (Table 2 and Figure 5 d). The most stable cyclones appear to be in the center of the LB, probably surrounding the LBE, having enhanced radius (Figure

7 b), slightly enhanced orbital velocity (Figure 7 d) and slightly decreased ellipse linearity (Figure 7 f).



### 3.3 Mean drift pattern of eddies and the ambient flow

The size, shape, nonlinearity, lifetime and generation sites of eddies influence the properties and fate of water masses trapped inside the eddies. How their water properties change with time can be related to processes within the eddies, how they drift, interactions with other eddies or the ambient flow, or how they are affected by the atmospheric forcing. Here and in Section

3.4, we study changes along drifter trajectories to investigate the drift of eddies as well as the evolution of their water masses with time. We compare this with the ambient flow (the AM drifters). We obtain the mean drift of eddies and the ambient flow (Figure 8) by averaging all velocities into longitude and latitude bins. We show results at 200 m to be consistent with earlier. Eastward and northward velocities are computed along ridges from the rate of change of position using the lonres and latres variables (Table 1), which give the position of the mass center of the eddy, and for the AM drifters using their longitude and

latitude data. The velocity fields from ridges give the mean eddy drift (Figure 8 b,c). We also obtain the residual eddy drift by subtracting the Eulerian mean flow obtained from the ROMS model given in Figure 8 a, for both cyclonic (Figure 8 e) and anticyclonic ridges (Figure 8 f), as well as for the AM drifters (Figure 8 d). Note that the velocity fields are derived from the 2D drifters to ensure that they are at fixed levels (200 m).

The AM drifters on average follow the Eulerian mean flow from the ROMS model (i.e., residual drift of the AM flow is

small) (Figure 8 d). The cyclones and anticyclones exhibit a fairly similar drift, with smaller magnitude than the Eulerian mean flow (Figure 8 b,c). This results in large magnitudes of the residual eddy drift for both cyclones (Figure 8 e) and anticyclones (Figure 8 f). The slower drift of the cyclones and anticyclones is particularly pronounced on the slope with residual currents of 15-20 cm/s pointing southwards against the direction of the mean flow. Water masses in eddies therefore tend to experience increased residence time in the LB region and a longer transit time towards the Arctic, possibly leading to larger changes in

water mass properties compared to the ambient flow (further discussed in Section 3.4). We observe that the mean flow is weaker towards deeper levels (500 m) compared to 15 and 200 m (not shown). However, the eddy drift speeds have a fairly similar pattern at all levels. This indicates that the eddies at 500 m drift with a speed similar to the mean flow, but at 15 m, the eddy drift is relatively slower.

### 3.4 Evolution of water masses in eddies and in the ambient flow

The AW modification is particularly strong in regions with large eddy activity, such as the LBE and the secondary EKE maximum in the southeast of the LB (Bosse et al., 2018; Rossby et al., 2009a). In this section, we attempt to quantify the evolution of temperature, density and vertical displacements along ridges and AM drifters, to estimate the property rate of change of a water parcel inside an eddy compared to the ambient flow.

We first estimate the characteristic temperature anomalies for the cyclonic and anticyclonic ridges and compare these to

AM drifters. To remove the spatial and seasonal variability embedded in the drifter temperatures, we compute a background temperature climatology from the ROMS model by taking seasonal averages of temperature for winter (Jan-Mar), spring (Apr-Jun), summer (Jul-Sep) and fall (Oct-Dec) between 1996-2000, the same period as the Lagrangian simulations. These are interpolated onto the drifter trajectories and subtracted from the temperatures to give temperature anomalies. The PDFs



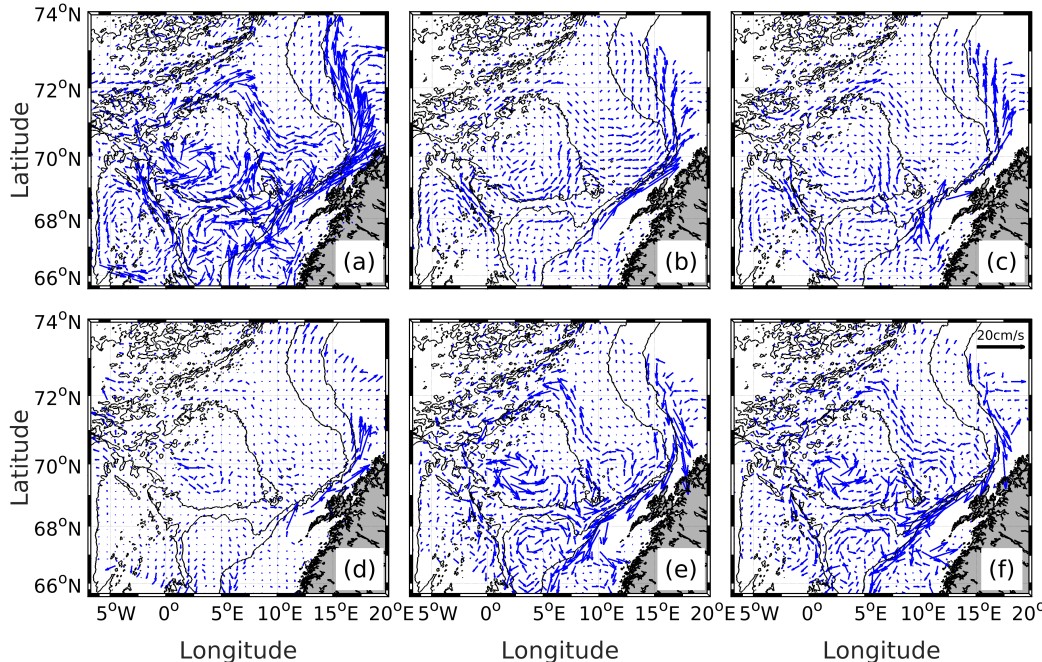

**Figure 8.** Averaged velocity fields at 200 m from ROMS or 2D drifters. (a) Eulerian mean flow from the ROMS model at 200 m averaged between 1996 and 2000; (b) drift of cyclonic ridges; (c) drift of anticyclonic ridges; residual flow from (d) AM drifters, (e) cyclonic ridges and (f) anticyclonic ridges. Residuals are obtained by removing the Eulerian mean flow (panel a). Scale of the arrows is given in upper right in panel f and is the same for all panels

of anomalies from the 3D drifters (Figure 9) indicate that anticyclones are warm and cyclones are cold compared to the

background flow, consistent with results obtained from ARGO floats (Raj et al., 2016). The PDFs are also centered around larger positive and negative values with depth for anticyclones and cyclones respectively (mean of $0.19^oC$ and $0.37^oC$ at 15 and 500 m respectively for anticyclones, and mean of $-0.25^oC$ and $-0.33^oC$ at 15 and 500 m respectively for cyclones), consistent with results obtained from hydrography (Sandalyuk et al., 2020). For the ambient flow (AM drifters), the temperature anomalies are centered around zero.

Warm and cold temperature anomalies for anticyclones and cyclones, respectively, can impact the cooling and warming experienced by water masses inside them. Since cooling and warming is related to an increase or decrease in density, this may also affect the vertical motion of the water masses. We therefore compute daily temperature changes and vertical displacements along drifter trajectories, and assign these values to the drifter's mean position that day. With a lifetime of one year this is 364 data points for each drifter or less if a drifter runs aground or exits the domain earlier. The daily temperature changes and the

vertical displacements for 3D drifters are then binned as before.





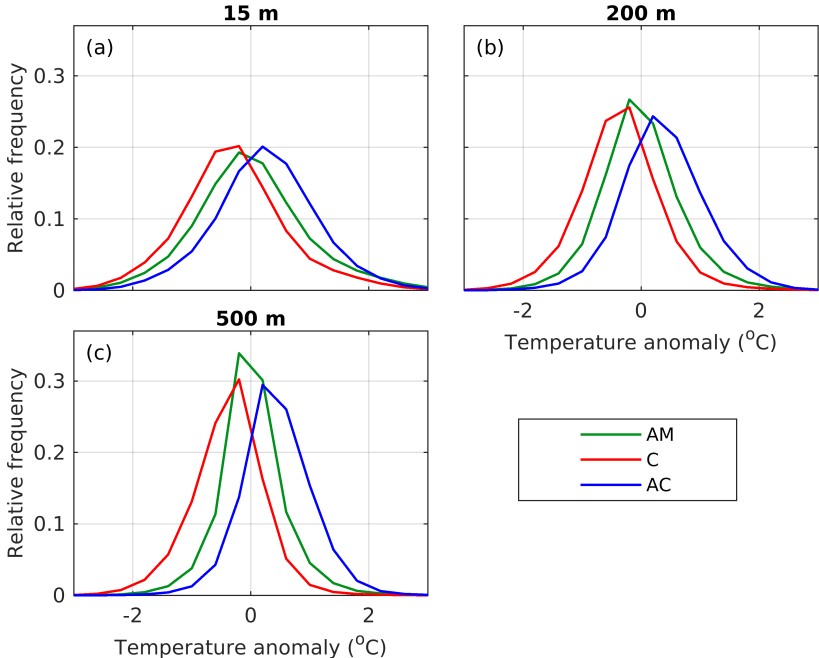

**Figure 9.** PDF distribution of temperature anomalies for 3D drifters deployed at (a) 15 m, (b) 200 m and (c) 500 m. The anomalies are relative to a seasonal background climatology from the ROMS fields (see text). PDFs are shown for AM drifters (AM, green), cyclonic ridges (C, red) and anticyclonic ridges (AC, blue).

For 3D drifters deployed at 15 m (Figure 10) there is an overall net cooling and sinking in the domain, with the largest values for the anticyclonic ridges (e,f). Along the cyclonic ridges (c,d) the drifters experience some warming (0.01-0.02 $^{o}$C/day) and upward motion (2-4 m/day) by the slope, but cooling in the basin (0-0.01 $^{o}$C/day), likely reflecting the atmospheric cooling close to the surface. For the 3D drifters deployed at 500 m (Figure 11) there is typically enhanced cooling and downwelling

in the anticyclones and enhanced warming and upwelling in the cyclones. However, while there is a fairly consistent pattern of cooling (anticyclones) and warming (cyclones) in the domain, the vertical displacements vary geographically. For the AM drifters the temperature and depth changes are generally small, except at the surface where they lose temperature and sink in response to atmospheric cooling.

There is no obvious relation between the temperature changes and the vertical motion in the different flow categories (Figure

10 and 11), i.e., we cannot say that a a change in temperature leads to a change in the vertical displacement. Note also that the results are based on daily differences and should be interpreted with caution. For instance, a water parcel entering the LBE from a colder environment could initially experience warming, but over longer time scales water masses in the LBE are generally cooled. It is therefore of interest to study how water masses in eddies and in the ambient flow change over a longer period of time. Since the 2D drifters remain at their deployment depth, a comparison with the 3D drifters provides information

about the effect of the temperature change on vertical motion or vice versa. We focus on the LB and compute time series of




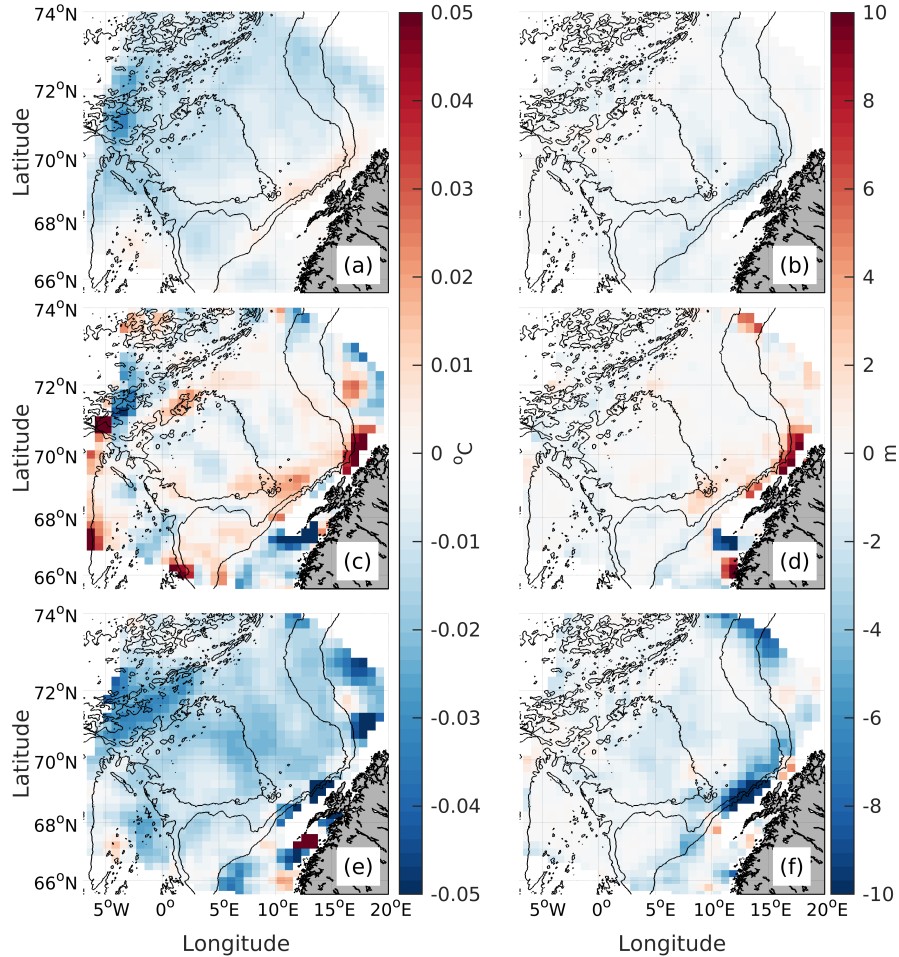

**Figure 10.** Averaged (a,c,e) daily temperature changes and (b,d,f) daily vertical displacements (colors) computed for (a,b) AM drifters, (c,d) cyclonic ridges and (e,f) anticyclonic ridges for 3D drifters deployed at 15 m. The maps are smoothed using a 9-point mean filter to remove some noise.

the temperature and density change and vertical displacements along all drifter trajectories inside the basin. Using all drifters that interacted with the LB (either deployed there or entered at a later time), we calculate the temperature differences $T_i - T_1$ (and density differences and vertical displacements similarly) for all trajectories while they are in the basin. $T_i$ and $T_1$ denote the temperature at index i and index 1 along a drifter trajectory after it entered or was deployed in the LB. If a drifter crosses
the basin boundary multiple times, each period in the basin is considered separately. This gives a record of how the properties change with time for each drifter segment inside the basin. Averaging over all drifter segments studied for every time step, we obtain a time series of the mean property change in the basin. This procedure is done for cyclonic and anticyclonic ridges, and



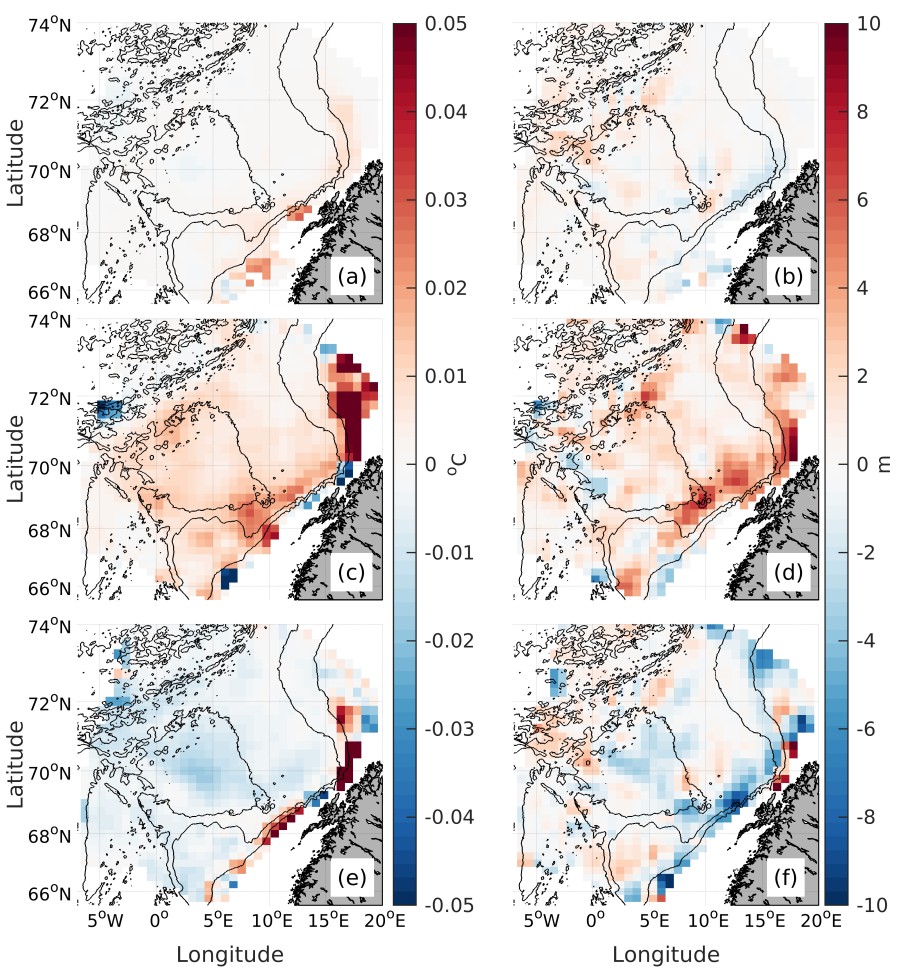

**Figure 11.** As in Figure 10, but for 3D drifters deployed at 500 m.

for AM drifters. To look at whether the LB is important, we also perform a similar analysis for comparison, using all drifters
in the entire domain and computing differences along all trajectories from deployment to termination.

The vertical displacements indicate a net sinking in the domain for the ambient flow and this is even more enhanced in
anticyclones (Figure 12). Water masses in cyclones on average stay at fixed depths with time, but there is some upward motion
at 500 m. The mean sinking is more pronounced in the LB compared to the full domain. Due to the initially warm signature of
anticyclones, we observe a net cooling within these (and the opposite for cyclones), that again is mainly enhanced in the LB
(Figure 12 e,f). Cooling is also experienced by the AM drifters, but this is most pronounced close to surface, possibly due to

a stronger atmospheric cooling there. 2D and 3D results agree fairly well, but with some differences. 2D cyclonic ridges at 15
m cool strongly in the LB compared to the 3D cyclonic ridges. Since the 2D drifters are close to the surface during their entire
lifetime, they are exposed to atmospheric cooling for extended periods compared to 3D drifters that are vertically spread and

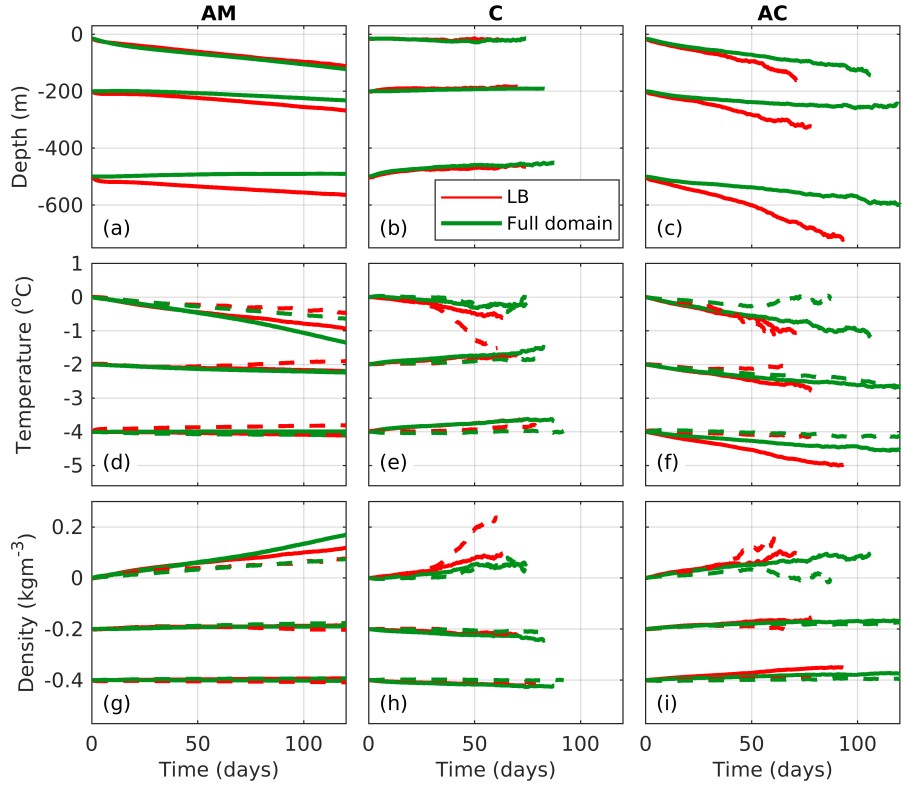

**Figure 12.** Time series of (a,b,c) mean vertical displacement for 3D drifters, (d,e,f) mean temperature change and (g,h,i) mean density change for 2D (dashed) and 3D (solid lines) drifters. Analyses for the LB (red) and the full domain (green) are shown for the (a,d,g) AM drifters, (b,e,h) cyclonic ridges (C) and (c,f,i) anticyclonic ridges (AC), and for different deployment depths (15m, 200 m and 500 m). To distinguish the drifters that were deployed at 15 m, 200 m and 500 m, an offset of -15 m, -200 m and -500 m is used for the vertical displacements, 0, -2$^o$C and -4$^o$C for the temperature changes and 0, 0.2 kgm$^{-3}$ and 0.4 kgm$^{-3}$ for the density. Note that the mean is based on fewer data points with increasing time and we therefore stop the time series when the mean is based on fewer than 100 data points.

can enter a cyclone at deeper levels. Also, for the anticyclones the 3D drifters deployed at 200 and 500 m experience more cooling and density increase than the 2D drifters, especially in the LB.

Cooling of the water parcels is typically accompanied by an increase in density (Figure 12 g,h,i). In these cases, a net sinking also often occurs, for example for water masses in anticyclones in the LB at 200 and 500 m. However, there are also examples of sinking, not associated with cooling or increase in density, for instance for AM 3D drifters in the LB deployed at 500 m (Figure 12 a,d,g). In this case the vertical motion could be related to movement along isopycnals that on average deepen due to weak stratification in the LB (e.g., Bosse et al. (2018)). The vertical motion of the 3D drifters is discussed further in Section

365    4.2.





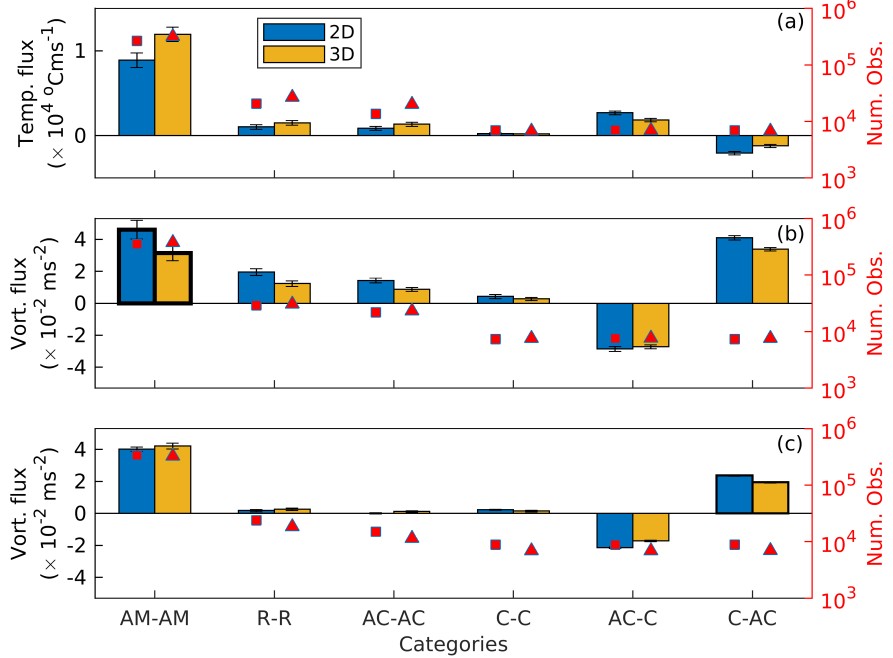

**Figure 13.** Estimates of the (a) net temperature flux and (b,c) net vorticity flux into the LB for (blue) 2D and (yellow) 3D drifters deployed at (a,b) 15 m and (c) 500 m depth, given on the left axes. The results are shown for selected entry/exit pair categories described in Section 3.5. Black errorbars show ± 3 standard deviations. Red squares and triangles show the number of drifter pairs used for the calculation for the 2D and 3D simulations respectively and are given on the right axes with a log scale. The abbreviations are: AM-AM=ambient flow in-ambient flow out, R-R=ridge in-ridge out (independent of whether it is an anticyclonic or cyclonic ridge), AC-AC=anticyclone in-anticyclone out, C-C=cyclone in-cyclone out, AC-C=anticyclone in-cyclone out, C-AC=cyclone in-anticyclone out.

### 3.5 Temperature and vorticity fluxes into the LB

The temperature and density changes are stronger in the LB compared to the full domain (Figure 12), consistent with earlier literature indicating that the LB is an important region for heat loss to atmosphere and modification of AW (Richards and Straneo, 2015; Dugstad et al., 2019a; Bosse et al., 2018; Rossby et al., 2009a). The cooling rate in the anticyclones (experienced 370 by drifters) are enhanced (Figure 10, 11 and 12), and we expect that water parcels in anticyclones will contribute to large heat fluxes into the LB. However, the small fraction of ridge points suggests that, summed over all water parcels, the total contribution from the ambient flow could be substantial. The relative contribution of vortices and ambient flow can be quantified by computing the net temperature and vorticity fluxes into the LB. Spall (2010) suggested that filaments can be important, e.g., give positive vorticity fluxes into the basin. In our analysis, ridges are a proxy for coherent vortices (Section 1) and AM drifters 375 contain the mean flow as well as filaments.

The net heat flux (or heat flux convergence) into the basin has earlier been computed to be positive via an Eulerian approach from observations (Segtnan et al., 2011) and from model simulations (Dugstad et al., 2019a). Here, we use a Lagrangian





approach to approximate this. Note that we use "temperature flux" (not heat flux) since we cannot integrate over a mass-conserving volume (as in the Eulerian sense). However, we interpret each drifter to carry a given mass, and by constraining an

equal number of entries and exits into the basin, we approximately conserve mass. The net temperature flux and vorticity flux into the basin are computed similarly. We search through all drifter trajectories and find all crossings with the LB boundary, and then tag them as entries or exits. For each entry/exit we obtain the values of temperature ($T$) and velocity $\mathbf{u} = (u, v)$ and estimate the temperature flux into the basin TF$= \mathbf{u}T \cdot \mathbf{n}$, where $\mathbf{n}$ is the local normal vector to the basin contour (pointing inwards so that entries are positive). Each entry/exit is then categorized as a cyclonic ridge, an anticyclonic ridge or ambient

flow. Thus, we obtain a data series of temperature fluxes TF across the LB boundary for all entries and exits for each of the three categories. To compute the net temperature flux, we subtract the temperature flux out of the basin (exits) from the temperature flux into the basin (entries), and then sum these differences for different groups. When performing the subtraction, the number of entries and exits are not necessarily the same. In order to keep an approximate mass balance, we proceed in the following way: Suppose that 1000 drifters enter the basin as ridges and 900 exit as ridges. To compute the contribution to the

net temperature flux from the ridges we can define a maximum of 900 entries/exits. To account for the spreading of the data we randomly subsample 75% of the 900 entries/exits. We then compute differences of 675 ($= 900 \times 0.75$) randomly chosen drifter pairs, and sum. This procedure is repeated 1000 times in a bootstrapping routine in order to get a mean and standard deviation of the net temperature flux from the ridges. It is of interest to compare the net temperature fluxes from the ambient flow (AM drifters) with the temperature fluxes from eddies (the ridges), and we again categorize the ridges into cyclones and

anticyclones. We compute the net temperature flux after computing differences for the ambient flow in - ambient flow out (AM-AM), ridge in - ridge out (R-R), anticyclonic ridge in - anticyclonic ridge out (AC-AC), cyclonic ridge in - cyclonic ridge out (C-C), anticyclonic ridge in - cyclonic ridge out (AC-C) and cyclonic ridge in - anticyclonic ridge out (C-AC). Of the detected entries/exits, about 300,000-400,000 are AM, 20,000-30,000 are ridges, 12,000-23,000 are AC and 7,000-9,000 are C for every depth and simulation. It is of interest to see whether the vertical motion of the 3D drifters prior to the crossings with

the LB boundary has an impact on the fluxes and we therefore include both 2D and 3D drifters in this analysis. Results for the net temperature flux is shown for 2D and 3D drifters deployed at 15 m (Figure 13 a), and the net vorticity flux is shown for 2D and 3D drifters deployed at 15 and 500 m (Figure 13 b,c respectively). Net temperature fluxes are similar at other levels, while net vorticity fluxes from drifters at 200 m are similar to results from 15 m. Results for AM-R or R-AM groups are similar to the AC-C and C-AC groups, but with smaller magnitudes and are therefore not shown.

Our analysis does not follow each individual drifter, but rather compares entries and exits. However, the procedure is statistically representative because we compare a large number of entries and exits (about 400,000-500,000) that are chosen randomly around the basin. Following the identity of each drifter would reduce the statistical significance as the number of samples would reduce drastically. For instance, in the AC-C group we would require that the drifter should enter in an anticyclone and exit in a cyclone. This happens very rarely (about 250 out of 500,000 cases and similar for C-AC). In quantifying heat brought in vs

heat brought out, these extra constraints would worsen the statistics and hamper our interpretation.

Summed over all categories, the net temperature flux into the basin is positive (Figure 13 a). The relative distribution between the groups clearly indicate that the AM drifters dominate the net temperature flux into the basin, and 2D and 3D results show the





same pattern. This suggests that the integral effect of the ambient flow is more important than eddies for the net temperature
flux. The net vorticity flux into the basin is similarly dominated by the AM drifters, especially for drifters deployed at 15
m (Figure 13 b) and 200 m (not shown). Considering the ridge categories, the largest magnitudes are found for the AC-C
and C-AC categories, but with opposite signs for both the temperature and vorticity fluxes. Consistent with the positive and
negative temperature anomalies for anticyclones and cyclones respectively (Figure 9), anticyclones bring warm water into the
LB and cyclones bring cold water out (resulting in relatively large values of AC-C). But anticyclones can also bring warm
water out of the LB and cold water in, also resulting in relatively large negative values of C-AC. A similar behavior goes for
the vorticity flux. The anticyclones bring negative vorticity in while cyclones carry positive vorticity out, resulting in relatively
large magnitudes for AC-C. Values for C-AC are similar, but with opposite sign since cyclones also bring positive vorticity in
while anticyclones carry negative vorticity out.

The number of observations is also given in Figure 13. The dominance of the AM-AM category is clearly related to the larger
number of available drifter pairs for the calculations (given that the net temperature or vorticity fluxes are more frequently
positive for AM-AM, summing up over much larger number of observations increases the positive total). The net temperature
fluxes into the basin *per drifter pair* are $(0.02 - 0.04)$ $^o$C ms$^{-1}$ for AM-AM and $(0.03 - 0.09)$ $^o$C ms$^{-1}$ for R-R, and vorticity
fluxes are $(0.01 - 0.09) \times 10^{-5}$ ms$^{-2}$ for both AM-AM and R-R, for all depths and simulations. Across the cyclonic and
anticyclonic ridge categories, temperature fluxes are AC-C=$(0.17 - 0.44)$ $^o$C ms$^{-1}$ and C-AC=$-(0.11 - 0.34)$ $^o$C ms$^{-1}$, and
vorticity fluxes are AC-C=$-(0.24 - 0.40) \times 10^{-5}$ ms$^{-2}$ and C-AC=$(0.27 - 0.59) \times 10^{-5}$ ms$^{-2}$ for all depths and simulations.
Both AC-AC and C-C categories show a positive temperature and vorticity flux but with smaller magnitudes (similar to R-R).
The results indicate that on average a water parcel in an eddy contributes with a larger positive net temperature flux to the
LB than a water parcel in the ambient flow. The AC-C and C-AC groups dominate, having opposite signs, but with the largest
values of AC-C. For the vorticity fluxes the AM-AM and R-R categories are similar. On average, a water parcel in an eddy
gives approximately the same vorticity flux into the basin as a water parcel in the ambient flow. But here larger magnitudes for
the AC-C and C-AC groups again indicate that a water parcel in an anticyclone or cyclone has a larger effect on the net vorticity
flux to the basin. The results are thus sensitive to the choice of studying the net fluxes over all drifter pairs or per drifter pair.
We discuss this further in Section 4.1.

Recall that the ambient flow includes mean flow, filaments and other submesoscale features. Spall (2010) found that filaments
often could carry large cyclonic vorticity (sometimes with $\zeta = 2f$) and suggested that these could give large fluxes into the
basin. We hypothesize that the large fluxes into the basin from the ambient flow in our study are related to filaments. The PDF
distribution of the net vorticity fluxes into the basin (not shown) computed from the differences for every entry/exit drifter pair,
is skewed towards positive values with a skewness of (0.93,0.67), (0.74,0.71) and (0.48,1.10) in the order (2D,3D) for AM
drifters deployed at 15 m, 200 m and 500 m, respectively. The relative vorticity is positively skewed for both entries and exits
of AM drifters, but the skewness is larger for the entries. In fact, for the 2D drifters at 15 m, 6,165 entries have relative vorticity
larger than $\frac{f}{2}$, and only 15 entries smaller than $-\frac{f}{2}$. 3,426 exits have relative vorticity larger than $\frac{f}{2}$ and 22 exits smaller than
$-\frac{f}{2}$, clearly indicating that the entries bring more positive vorticity into the basin than the exits carry out of the basin.



As a proxy for the contribution by filaments, we estimate the relative contribution of these entries/exits, associated with large absolute values of relative vorticity ($|\zeta| > \frac{f}{2}$), to the total net vorticity flux into the basin given as the AM-AM category in Figure 13 b,c. Due to more entries (6,165+15=6,180) than exits (3,426+22=3,448), the remaining 2,732 exits are randomly picked to get a mass balance before computing the differences, and this is repeated 1000 times in a bootstrapping routine. We use 75% of the data to be consistent with earlier analysis. The differences are then computed and summed. Spreading of the data is small and we estimate that 18-19% of the net vorticity flux come from the entries/exits associated with large vorticity. The same calculation for other depths and the 3D simulation gives about 1% and 0% at 200 and 500 m depth for the 2D drifters respectively, and 7%, 4% and 3% for the 3D drifters deployed at 15 m, 200 m and 500 m respectively. Thus, the water parcels associated with strong cyclonic vorticity, that are not in eddies, can give large contributions to the vorticity budget in the basin at the surface. Note the difference between 2D and 3D drifters deployed at 15 m. Large values of relative vorticity ($|\zeta| > \frac{f}{2}$) are traced much more frequently by the 2D drifters that are close to surface than the 3D drifters that are more spread in the vertical. Filaments carrying strong relative vorticity therefore mainly appear to be a surface feature. A similar calculation for the temperature fluxes gives percentages of about 13% 0% and 0% for the 2D drifters at 15 m, 200 m and 500 m respectively, and 2%, 1% and 1% for the 3D drifters deployed at 15 m, 200 m and 500 m respectively. Again the 2D drifters at 15 m shows largest results. Note that for these drifters the temperature fluxes and vorticity fluxes computed in Figure 13 a,b are computed from 270,000 drifter pairs while the results for large relative vorticity are based on $6,180 \times 0.75 = 4,635$ drifter pairs. Thus, this small subset of about 2% of the drifter pairs accounts for about 19% of the net vorticity flux and 13 % of the net temperature flux into the basin. Filaments therefore seem to play a significant role for the heat and vorticity budgets at the surface.

## 4  Discussion

### 4.1  Eddy sampling by the drifter trajectories

The relative small fraction of ridge points (6%) in the drifter trajectories suggests that drifters spend most of their time outside of coherent eddies. A comparison of the Lagrangian and Eulerian OW maps (Figure 4 a,b) also indicates that the fraction of drifter data points sampling OW<0 are generally smaller than the fraction of grid cells in the model that have OW<0 (Section 3.1). The two estimates are, however, similar early in the deployment, suggesting that uniformly-deployed drifters (that could be deployed in eddies or the ambient flow) reflect the Eulerian OW field of the model in the beginning (Figure 4 c). With time, the fraction of drifters sampling OW<0 rapidly decreases and after 15-20 days it stabilizes around 34-35%. This may imply that the drifters tend to follow the rim of eddies and reach a barrier that prevents them from entering the core of the eddies. Strong gradients in potential vorticity (PV) between eddies and the ambient flow can impose such barriers throughout most of the year (possibly except during winter). This has been shown, for example, for the LBE from cruise data and RAFOS floats (Bosse et al., 2019). The PV gradients likely constrain the number of drifters that enter eddies. The anticyclonic eddies in this study have a longer lifetime and a more circular shape reflecting nonlinear motion. The PV gradients are therefore likely stronger for the anticyclones than the cyclones, possibly explaining why the fraction of drifters experiencing negative vorticity is smaller than that with positive vorticity (Figure 4 c). As for the initial adjustment, one possible explanation may be tied to the



secondary circulation within coherent vortices. In particular, Bashmachnikov et al. (2018) found that such circulation within the LBE consisted of a divergent horizontal flow in and above the vortex core. Assuming such secondary circulation occurs in other anticyclones, and that the flow pattern is opposite for cyclones, this effect could result in an initial drop of the OW<0 fraction with negative vorticity sampled by drifters, and the initial increase for the positive vorticity.

The above discussion may seem contradictory to the results obtained from the MWRA routine. Studying the OW parameter
and the relative vorticity, we find more drifters experiencing OW<0 with cyclonic vorticity. In contrast, the MWRA routine detects more anticyclonic ridges, a result which is consistent with findings reported by Raj et al. (2016) and Volkov et al. (2015). Here it is worth remembering that the criterion of OW< 0 is not thought to be sufficient for identifying coherent vortices. So the fraction of positive and negative vorticity, even where OW< 0, likely also reflect other flow features. In particular, the surrounding flow around anticyclones is typically dominated by cyclonic filamentary structures (not shown, but see Spall,
2010). And since it appears that drifters in our simulations may be constrained by the rim of eddies, the larger amount of anticyclonic ridges could therefore imply that drifters sample more filaments with cyclonic vorticity.

The amount of long-lived coherent eddies in the model is clearly smaller than what is suggested by the OW< 0 criterion alone. For instance, the fraction of drifter data points that sample OW< 0 for more than 1 day or 2 days decreases drastically compared to a detection based on instantaneous OW values (Figure 4 c). Putting additional constraints to eddies computed
from the ROMS model would likely lead to an additional decrease of the computed fractions. It is therefore not surprising that the fraction of ridge points (black solid line in Figure 4 c) is about 6%. So a dominance of the AM-AM category in the total temperature and vorticity fluxes into the basin (Figure 13) may in fact be reasonable. One could argue that if the fraction shown for the OW parameter (Figure 4 c, green solid line), which is about 6 times larger than the fraction of ridge points (Figure 4 c, solid black line) was a true estimate of the fraction of eddies, the bars given for all categories except AM-AM in Figure 13
would have about 6 times larger magnitude. But note that as the results of the R-R category are based on about 30,000 drifter pairs while the AM-AM category is based on about 400,000 drifter pairs a six-fold increase of R-R (giving 180,000 drifter pairs) would lead to a decrease to about 5/8 of the results shown for the AM-AM category (which would then include 250,000 drifter pairs). In this case the net temperature flux into the basin would be dominated by the ridge categories, but the AM-AM category would still play an important role, and even more so for the vorticity flux. Since using OW<0 alone as a criterion for
coherent eddies is unrealistic and we expect the actual fraction of eddies to be more similar to the estimates from the MWRA routine, we conclude that the ambient flow is important for a balanced heat and vorticity budget.

The most sensitive parameter choice in the MWRA routine is the choice of minimum ridge length (RL). For the above analysis we used a minimum length of 3.1 cycles (Section 2.3), but as a sensitivity experiment we ran the MWRA routine with the RL set to 1.6 cycles, similar to Lilly et al. (2011). For this test we only used 10,000 randomly chosen drifters to reduce
computation time. The shorter minimum ridge length resulted in 65-70% more ridge point detections (Figure 4 c, other depths and simulations show similar results). But we note that RL=1.6 is similar to the number of oscillations within the wavelet and therefore on the limit of a meaningful eddy signal. As the wavelet transform is a joint function of the wavelet and the trajectory signal, noise or smaller jumps in the trajectories could lead to copies of the wavelet itself in the wavelet transform (Lilly and Olhede, 2009; Lilly et al., 2011). Hence, the sensitivity experiment discussed here should be interpreted with caution.



We conclude that the number of eddies identified using the MWRA routine could be underestimated. However, using OW<0
as a proxy for eddies in the ROMS simulation will definitely lead to an overestimate and is not realistic (Trodahl and Isachsen,
2018; Penven et al., 2005; Raj et al., 2016). We propose that our analysis statistically captures the relative contribution of eddies
and the ambient flow, and highlights the importance of the ambient flow for the net temperature flux and vorticity flux.

### 4.2    Change of water properties in eddies and the ambient flow

The eastern Nordic Seas that we study here are primarily temperature-stratified. We hence expect that water parcels which
are cooled will eventually sink from gravitational adjustment. An indication that this process is taking place place near the
surface, where parcels are directly exposed to air-sea heat loss, is seen in Figure 12 for the 3D AM (ambient water) and AC
(anticyclones) drifter classes seeded at 15 m. 2D drifters seeded at the same depth generally experience smaller temperature
drops (and density increases) than do 3D drifters. A plausible interpretation is that the fixed-level drifters are not allowed
to gravitationally adjust along with the coldest waters that they encounter. Vertical motion of 3D drifters that coincide with
changes in temperature and density which are larger than for 2D drifters are also seen at depth and in particular for drifters
in anticyclones in the LB. However, examples of vertical motion not accompanied by changes in temperature or density also
occur, for instance for the AM drifters in the LB deployed at 200 and 500 m. The vertical motion experienced by these drifters
in particular, but partly also the other drifter classes, may reflect adiabatic movement along sloping isopycnals. The pronounced
sinking experienced by 3D AM drifters at depth in the LB compared to the full domain can partly be related to the deepening
of isopycnals observed there (Rossby et al., 2009a; Bosse et al., 2018).

The most pronounced signal seen in Figures 10, 11 and 12 is perhaps the asymmetry in what drifters experience in cyclones
and anticyclones. Drifters trapped in anticyclones mostly experience downward motion while those in cyclones mostly expe-
rience upward motion (drifters in cyclones don't sink near the surface like AM and AC drifters do). The vertical movement
here too may be reflecting gravitational adjustment resulting from the cyclones gradually warming up (as they are anomalously
cold) and the anticyclones gradually cooling. But some of the asymmetry may also be related to the secondary circulation
within vortices mentioned above. Bashmachnikov et al. (2018) found from model studies using MITgcm that vertical motion
in the LBE has a complex structure with a lateral divergence of water masses at upper levels, leading to upward motion in and
above the eddy core. This is compensated by downward motion along the flanks of the vortex. If such a secondary circulation
pattern is common for anticyclones in the LB—and opposite for cyclones—then our drifter statistics should be affected sys-
tematically. We deploy drifters uniformly, thereby sometimes in the eddy cores but most often outside eddies (since the eddies
cover a smaller portion of the domain than the ambient flow). Although some drifters occasionally enter eddy cores, our results
indicate that they spend a larger fraction of their time on ridges outside the cores. In particular, we found that ridges in the LBE
typically trace out radii that are significantly larger than the core radius reported by Fer et al. (2018) and Søiland and Rossby
(2013), implying that these drifters were often looping on the eddy flanks. We therefore speculate that some of the observed
downwelling of drifters in anticyclones and upwelling in cyclones is related to the secondary circulation within the vortices. It
is worth noting that vertical advection by such secondary circulation through the stably-stratified water masses in and around
a vortex will lead to the kinds of temperature and density changes recorded by our synthetic 3D drifters. In steady state the





vertical advection must, in turn, be balanced by turbulent buoyancy fluxes. This requirement is at least indirectly supported by
observational reports of enhanced turbulent dissipation levels within the LBE (Fer et al., 2018).

So the observed vertical motion of water masses in and around the LB can therefore be related to several processes. On the one hand, cooling and densification can cause subduction, but only if the water masses get heavier than their surroundings. The vertical motion can also be related to motion along sloping isopycnals. For eddies, a secondary vertical motion within the eddy may also occur. In the first and third case the water masses change their properties and exhibit water mass transformation
while the second case is adiabatic. Concerning actual water mass transformation, we propose that this takes place mainly near the surface but also in vortices at depth since here too the 2D and 3D temperature and density changes are different.

### 4.3    Vertical structure in the AW-LB exchange

We used $|\zeta| > \frac{f}{2}$ as a proxy for filaments in the ambient flow and found that such filaments give a positive net vorticity flux into the basin, consistent with the idealized model simulations of Spall (2010). More specifically, filaments gave significant
contributions to the LB vorticity and heat budgets at the surface (15 m), where the subset of about 2% of the 2D entries with $|\zeta| > \frac{f}{2}$ contributed to about 19% of the vorticity flux and about 13% of the temperature flux into the basin. The 3D drifters deployed at 15 m as well as drifters at deeper levels gave smaller contributions. This points to a more dynamically active surface than at deeper levels, likely caused by more variability due to wind and surface buoyancy fluxes. Furthermore, the eddies found by the MWRA routine were dominated by anticyclonic rotation possibly explaining why we on average found that filaments
were dominated by positive vorticity. The anticyclones and cyclones have a shorter lifetime at surface than deeper levels and are more elongated (Table 2 and Figure 5 d), implying a more unstable character. We speculate that variable wind fields disturb the generation of eddies and that enhanced diapycnal processes drain energy and limit the lifetime of vortices, but favor the occurrence of filaments.

The importance of filaments at the surface is consistent with large magnitudes in the net temperature and vorticity fluxes
to the LB for the ambient flow (Figure 13 a,b). At 500 m, the vorticity fluxes tied to AM drifter points are smaller (Figure 13 c, about 1/5 of the results at 15 and 200 m) and the contribution from filaments vanish. The same is also partly true for the temperature fluxes. Since the magnitudes associated with the ridge categories are similar for all depths, this implies a relatively larger contribution of fluxes related to vortices at deeper levels. Focusing on the temperature fluxes, earlier studies have suggested that divergence of eddy heat fluxes from the continental slope towards the basin interior dominate at subsurface
levels (Dugstad et al., 2019a). The eddies at depth have longer lifetimes (Table 2 and Figure 5) and about the same drift speed as eddies at shallower levels (not shown), meaning they are likely to propagate longer distances and reach the basin. Given that the anticyclones are warm and often detected close to the basin (not at the slope, Figure 6) these can therefore drift to the basin feeding the LB with warm water. Considering only the anticyclonic entries to the LB from drifters deployed at 500 m, about 52% of all anticyclonic entries from 2D drifters and 41% from 3D drifters occur along the southeastern part of the LB,
although this segment only accounts for about 15% of the total length of the LB 3000 m contour (between green triangles in Figure 1 a). The entries give temperature fluxes that account for about 71% and 54% of the total temperature fluxes from all 2D and 3D anticyclonic entries, respectively. We do not investigate the exits in this particular case and therefore do not consider





these numbers in a mass-conserving framework. However, since the ridge categories in Figure 13 (where mass is conserved) show positive temperature fluxes and have a relatively higher importance at 500 m, we propose that the anticyclonic entries

from the slope at depth (500 m) can give a significant contribution to the heat budget of the LB. The entries of anticyclones from the slope are consistent with earlier literature (Köhl, 2007; Volkov et al., 2015; Raj et al., 2015; Isachsen et al., 2012). In addition, the anticyclones at 500 m give relatively large temperature fluxes into the LB, consistent with Dugstad et al. (2019b) who found that the water masses at these depths that were cooled in the basin came mainly from the slope. The findings of Dugstad et al. (2019b) therefore appear to be related to anticyclonic eddies. We find that the anticyclones trace out a lateral

tongue of a warm temperatures intruding from the slope far into the LB (not shown). This is different than the temperature signals at shallower levels (Figure 1 b) showing warm temperatures also over the Vøring Plateau. We therefore propose that anticyclones generated close to the slope at deeper levels (about 500 m) can give significant contribution to the LB heat budget and that this contribution is more important at depth than at shallower levels where the ambient flow dominates.

## 5   Summary and conclusions

In this study we have investigated the eddy activity in the Norwegian Sea, with focus on the Lofoten Basin, using a Lagrangian framework. We used high-resolution model fields and analyzed about 200,000 2D and 3D synthetic drifter trajectories seeded at 15, 200 and 500 m. A multivariate wavelet ridge analysis (MWRA) was used to identify and characterize cyclonic or anti-cyclonic ridges (coherent vortices referred to as eddies). We quantified how the water properties in cyclonic and anticyclonic eddies, as well as in the ambient flow, evolved with time (i.e., changes in temperature and density) and studied the net temper-

ature and vorticity fluxes into the basin.

The drifters sampled larger radii for the anticyclones (20-22 km) compared to cyclones (17-19 km), longer lifetimes (16-24 days compared to 13-19 days) and a more circular shape (ellipse parameter $\lambda$ =0.45-0.50 compared to $\lambda$ =0.51-0.57), indicat-ing a more stable character for the anticyclones. Compared to the climatological background state, water masses in anticyclones were anomalously warm while water masses in cyclones were anomalously cold. Water masses in eddies experienced large

changes in water properties (e.g., temperature and density) and cooled at a rate of 0.02-0.04 $^{o}$C per day in anticyclones and warmed at a rate of 0.01-0.02$^{o}$C per day in cyclones. Furthermore, water masses in anticyclones mainly downwelled while water masses in cyclones mainly upwelled. By comparing 2D and 3D drifters, the vertical motion was hypothesized to be related to several processes such as water mass transformation via direct cooling or warming, or via a secondary circulation caused by divergence of water masses above the anticyclonic eddy cores (convergence for cyclones), leading to changes in

temperature and density. Also, the drift of eddies along inclined isopycnals could lead to vertical motion without any change of water properties. A water parcel in an eddy typically contributed more than the ambient flow to the net temperature and vorticity fluxes into the LB. However, due to a larger fraction of drifter data points that were not ridge points (about 94%), the total temperature fluxes and vorticity fluxes were dominated by the ambient flow when summed over all entry/exit drifter pairs.

The AM drifters (outside coherent eddies) could in some occasions exhibit very strong relative vorticity ($|\zeta| > 0.5f$). We

interpret these as filamentary structures surrounding eddies and found that this rather small subset of drifters could give a large
contribution to the temperature and vorticity fluxes into the basin at surface (about 13% and 19% respectively). We speculate that filaments around eddies therefore play a significant role for the vorticity and heat budgets near the surface. At deeper levels (500 m) coherent vortices become more important, and anticyclones, in particular, are important for bringing warm water into the LB from regions close to the slope.

Our study has used a novel Lagrangian method to detect and characterize coherent eddies in the Norwegian Sea, to compare the movement and transformation of water parcels in eddies and the ambient flow and also to assess the relative contributions to transport of heat and vorticity into the Lofoten Basin. The indication by the Lagrangian data that long-lived coherent eddies may be less prevalent than previously thought motivates further comparison with Eulerian studies. Furthermore, the possible role of filaments in the upper layers, the link between the generation of eddies on the slope and their exchange with the LB

at deeper layers and, finally, the possible link between vortex secondary circulation and the transformation of AW all warrant further studies.

*Data availability.* The ROMS model fields used to perform the Lagrangian simulations are available at the Thredds Service at the Norwegian Meteorological Institute (https://thredds.met.no/). The Lagrangian simulations will be archived open-access, at the Norstore research data archive (https://archive.norstore.no/)

*Author contributions.* JD performed the Lagrangian simulations, the multivariate wavelet ridge analysis, analyzed the data and wrote the paper. PE and IF provided ideas and discussions that helped interpreting results and forming the paper throughout the process.

*Competing interests.* Ilker Fer is a member of the editorial board of *Ocean Science*, but other than that the authors declare no competing interests.

*Acknowledgements.* This study received funding from the Research Council of Norway, through the project Water mass transformation
processes and vortex dynamics in the Lofoten Basin in the Norwegian Sea (ProVoLo, project 250784). The ROMS simulation was made by Marta Trodahl and Nils M. Kristensen of the Norwegian Meteorological Institute and run on resources provided by UNINETT Sigma2-The National Infrastructure for High Performance Computing and Data Storage (projects NN9431K and NS9431K). The drifter simulations were performed on servers provided by the Norwegian Meteorological Institute in Oslo, Norway. We thank Jonathan M. Lilly for help with running and interpreting results from the multivariate wavelet ridge analysis, and Knut-Frode Dagestad for the help with performing the Lagrangian
simulations.





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
