# Peer review of "The mesoscale eddy field in the Lofoten Basin from high-resolution Lagrangian simulations"

_Ocean Science, 2020_

## Referee Comment (RC1) · Sarah Gille (Referee) · 29 Nov 2020

This manuscript uses a novel approach to assess the contributions of eddies to the Lofoten Basin. In this study, Lagrangian particles are simulated numerically using the ROMS model and are diagnosed using multi-variate wavelet ridge analysis, an approach which allows the authors to readily identify the presence of coherent vortex like structures. The manuscript was written as part of the lead author's PhD thesis, which I also had the pleasure of reviewing, and in a second reading, I remain impressed by the effectiveness of the analysis approach and the clear delineation of contributions from anti-cyclonic vortices, cyclonic vortices, and ambient flow. The approach is effective,

the results are clearly presented, and the findings will be relevant to readers of *Ocean Science*.

I have considered the 15 criteria provided for *Ocean Science* reviewers, and on the whole, I think the manuscript is in excellent shape.

There are a few issues that should be addressed prior to publication.

First, the analysis in section 3.5 examines the net transport into and out of the Lofoten Basin due to ambient flow, anti-cyclonic vortices, and cyclonic vortices. Although the analysis approach is clever and original, I think that it runs the risk of over-interpreting effects. The analysis pairs separate trajectories for flow into the basin and flow out of the basin to consider the net impact on the basin. However, as the authors note, very few particles actually transition from being in an anti-cyclonic flow on entrance to a cyclonic flow on exit (or vice versa). If particles don't actually experience this change, then using a bootstrapping approach to assess the net contribution due to this unrealistic scenariio seems risky. The manuscript would be stronger if the authors simply examined the net flux into the domain from each of the categories of particles and then separately examined the net flux out of the domain from each category of particle. (Alternatively, if pairing particles at entrance with particles at exit seems imperative, then this should be done using single particles only, without randomly matching entrance particles with other exit particles.)

Related to this, at about line 390, the authors explain the use of a boostrapping routine to estimate the contributions of particles of different types to the net flux. It's not clear that a bootstrapping approach is necessarily needed for this. If the statistics are relatively Gaussian, then it should be sufficient to compute the mean temperature flux and the standard error of the mean, without needing to go through the computational effort to compute a large bootstrap sample. If bootstrapping is formally necessary, then a bit more explanation would help readers understand why.

Figures 5 and 9 are identified as probability density functions, but neither appears to be normalized so that area under the curve integrates to one. Either they should not be labeled as pdfs (perhaps "distributions of relative frequency"?) or the plotted curves should be normalized by bin width, so that integral of the area under the curve is one.

Figure 12 shows line plots that would be enhanced if statistical uncertainties could be added to the lines. This wold allow readers to judge when the LB region differs statistically from the full domain.

In Figure 2d, I'm used to seeing wavelet transforms shown with an envelope to indicate the range of validity. Is there an applicable envelope in this case?

There are a number of typos, and I will separately upload a commented version of the pdf, in which I have marked suggested edits.

Please also note the supplement to this comment:
https://os.copernicus.org/preprints/os-2020-103/os-2020-103-RC1-supplement.pdf

**Supplement:**

[revised manuscript text omitted]

---

## Referee Comment (RC2) · Stefanie Ypma (Referee) · 11 Dec 2020

This manuscript by Dugstad et al. presents a thorough analysis of the eddy field in the Lofoten Basin using a multi-variate wavelet ridge analysis. Doing so, they've increased the understanding of the formation regions and characteristics of anticyclonic and cyclonic eddies and their respective importance for the heat transport and water mass transformation in the basin. The approach is novel, the paper is very clearly structured and written and regarding the 15 criteria provided for *Ocean Science* reviewers, I agree with Sarah Gille that the paper is in excellent shape.

I would like to add three comments in addition to the issues already raised by Sarah

Gille that should be addressed prior to publication.

1. A discussion is missing on the sensitivity of your results to the spatial and temporal seeding distribution of the Lagrangian particles. You seed particles on a 40x40 rectangular grid, but what is the distance between two particles and how does that compare to the average radius of the eddies? In other words, how many particles generally reside in 1 eddy? Regarding the temporal scale, you only seed particles once every week. As you discuss that it is difficult for particles to 'enter' eddies due to high vorticity gradients, aren't you under-sampling the eddy field due to the relatively low seeding frequency?

2. You mention that you don't add any diffusivity, so the particle displacement is purely advective. As the Lofoten Basin is characterised by strong heat losses, there is quite some convection going on. How well can your particles describe vertical motions and temperature changes of water parcels if this convective behaviour is not included?

3. Some of the figures can be improved by adding more clear labels. Comments on the figures, and some other minor comments are marked in the supplement.

Please also note the supplement to this comment:
https://os.copernicus.org/preprints/os-2020-103/os-2020-103-RC2-supplement.pdf

**Supplement:**

[revised manuscript text omitted]

---

## Author Comment (AC1) · 23 Jan 2021

Author response to reviewer Sarah Gille's comments on "The mesoscale eddy field in the Lofoten Basin from high-resolution Lagrangian simulations" by Dugstad et al.

The reviewer's comments are given below in black Times New Roman font, with our response in red Arial font.

Reviewer 1– Sarah Gille

This manuscript uses a novel approach to assess the contributions of eddies to the Lofoten Basin. In this study, Lagrangian particles are simulated numerically using the ROMS model and are diagnosed using multivariate wavelet ridge analysis, an approach which allows the authors to readily identify the presence of coherent vortex like structures. The manuscript was written as part of the lead author's PhD thesis, which I also had the pleasure of reviewing, and in a second reading, I remain impressed by the effectiveness of the analysis approach and the clear delineation of contributions from anticyclonic vortices, cyclonic vortices, and ambient flow. The approach is effective, the results are clearly presented, and the findings will be relevant to readers of OceanScience.

I have considered the 15 criteria provided for Ocean Science reviewers, and on the whole, I think the manuscript is in excellent shape.

We thank you for your constructive comments. They have led to a much improved manuscript, including a completely new flux calculation and a new Figure 13. Re-considering the flux calculations has also lead us to reduce the discussion about 2D drifters, since 3D drifter trajectories are thought to be more representative of actual flow paths. We address your specific comments below.

There are a few issues that should be addressed prior to publication.

First, the analysis in section 3.5 examines the net transport into and out of the Lofoten Basin due to ambient flow, anticyclonic vortices, and cyclonic vortices. Although the analysis approach is clever and original, I think that it runs the risk of over-interpreting effects. The analysis pairs separate trajectories for flow into the basin and flow out of the basin to consider the net impact on the basin. However, as the authors note, very few particles actually transition from being in an anticyclonic flow on entrance to a cyclonic flow on exit (or vice versa). If particles don't actually experience this change, then using a bootstrapping approach to assess the net contribution due to this unrealistic scenario seems risky. The manuscript would be stronger if the authors simply examined the net flux into the domain from each of the categories of particles and then separately examined the net flux out of the domain from each category of particle. (Alternatively, if

pairing particles at entrance with particles at exit seems imperative, then this should be done using single particles only, without randomly matching entrance particles with other exit particles.)

Your comment here is certainly well received. We have thought quite a bit more about how to present and interpret these kinds of estimates. But we do believe it is important to present temperature and vorticity fluxes in an approximately mass-conserving framework. We have therefore avoided computing fluxes from the entries and exits separately. However, your point that very few drifters actually switch from e.g. anticyclones to cyclones is one we have addressed in the revised manuscript. In a revised calculation, we follow your second suggestion, and follow the identity of each drifter and compute net fluxes (flux in – flux out) for each of these.

Specifically, we now compute the temperature/vorticity fluxes from drifters for the following 6 categories:
- Drifters that enter in anticyclones but then exit as any of the three categories
- Drifters that enter in cyclones but exit as any category
- Drifters that enter with the ambient flow but exit as any category
- Drifters that enter as any category but exit in anticyclones
- Drifters that enter as any category but exit in cyclones
- Drifters that enter as any category but exit with the ambient flow.

The results are given in Figure 1 below - this will be a new Figure 13 in the manuscript. Please note that we will now use 'AF' for ambient flow, and not 'AM', due to comments from reviewer Stefanie Ypma. The results from this alternate calculation are largely in agreement with the previous calculation in that both heat and vorticity fluxes into the central Lofoten Basin appear to be dominated by the ambient flow.

We now show and discuss flux estimates only for 3D drifters, because these offer a better representation of real flow paths. For Figure 12 of the manuscript (see below) we compare 3D and 2D drifters to investigate the relation between vertical movement and temperature changes. However, retaining the 2D drifters in the discussion of fluxes does not add to the manuscript. As a result, we also omit reference to the 2D estimates when we later discuss the contribution of filaments. On that note, we also shorten the discussion on the specific role of filaments, particularly with reference to flow having vorticity larger than $f/2$. This threshold value was relatively arbitrary, and a thorough investigation into this will be deferred to later work. Here the revised text will largely limit the discussion to the observation that the large vorticity fluxes from the ambient flow category likely implies small scales, e.g. filaments, rather than large-scale mean flow.

[Figure]

*Figure 1: Temperature (a, c, e) and vorticity (b, d, f) fluxes for 3D drifters that are deployed at (a, b) 15 m, (c, d) 200 m and (e, f) 500 m. The number of observations are given with red triangles and are the same for both the temperature and vorticity fluxes. Error bars indicate twice the standard error of the mean. Abbreviations are: AFi=Ambient flow in, ACi=Anticyclones in, Ci=Cyclones in, AFo=Ambient flow out, ACo=Anticyclones out, Co= Cyclones out. Thick black edges on the AFi and AFo categories in panel (b) and (d) indicate that the bars are given as 1/3 of their actual size (for visualisation purposes).*

Related to this, at about line 390, the authors explain the use of a boostrapping routine to estimate the contributions of particles of different types to the net flux. It's not clear that a bootstrapping approach is necessarily needed for this. If the statistics are relatively Gaussian, then it should be sufficient to compute the mean temperature flux and the standard error of the mean, without needing to go through the computational effort to compute a large bootstrap sample. If bootstrapping is formally necessary, then a bit more explanation would help readers understand why.

We agree with this comment and also note that choosing 75% of the drifters for each iteration of the bootstrapping was also a fairly random choice. We therefore decided to replace this form of uncertainty estimate with a more classical one. Note that our drifters are deployed every week over three years (1996, 1997 and 1998) with roughly the same amount of drifters deployed each year. Instead of estimating the total temperature and vorticity fluxes through all

years, we now split the drifters into three groups based on which year they are deployed. We thereby get three independent estimates for the net temperature/vorticity fluxes (from drifters deployed in 1996, 1997 and 1998). In our new Figure 13 (Figure 1 above) we show the 3-sample mean and also twice the standard errors (as a 95% confidence interval for the mean). Through this procedure, the relative importance between ambient flow and eddies is still clear.

Figures 5 and 9 are identified as probability density functions, but neither appears to be normalized so that area under the curve integrates to one. Either they should not be labeled as pdfs (perhaps "distributions of relative frequency"?) or the plotted curves should be normalized by bin width, so that integral of the area under the curve is one. Agreed. We decided to keep the curves but to refer to these as "distributions of relative frequency" (with "PDFs" changed to "DRFs", accordingly).

Figure 12 shows line plots that would be enhanced if statistical uncertainties could be added to the lines. This wold allow readers to judge when the LB region differs statistically from the full domain. Thanks for the input. We have now computed the standard error and included these as vertical error bars. To be consistent with Figure 13, the error bars here also show twice of the standard error (to indicate a 95% confidence interval for the mean). We first tried to plot the standard error as a shadow in the background but had to abandon this approach due to a large number of curves that made the figure crowded. Therefore, we have computed and plotted the error bars for specific days (30 and 60) after the drifters entered the basin. To distinguish results from the basin (red) and the full domain (green), error bars are plotted with an offset of +2 days for red and -2 days for green. For better visibility, we show error bars only for 3D results in an attempt to keep the figures simple. Note that error bars are computed for all categories, but due to their small magnitudes they are hardly visible for the ambient flow. The new figure will look like the following:

[Figure]

*Figure 2: Time series of (a, b, c) mean vertical displacement for 3D drifters, (d, e, f) mean temperature change  and (g, h, i) mean density change for 2D (dashed) and 3D (solid lines) drifters. Analyses for the LB (red) and the full domain (green) are shown for the (a, d, g) AF drifters, (b, e, h) cyclonic ridges (C) and (c, f, i) anticyclonic ridges (AC), and for different deployment depths (15m, 200 m and 500 m). To distinguish the drifters that were deployed at 15 m, 200 m and 500 m, an offset of -15 m, -200 m and -500 m is used for the vertical displacements, 0, -2ºC and -4ºC for the temperature changes and 0, 0.2 kg m⁻³ and 0.4 kg m⁻³ for the density. The mean is based on fewer data points with increasing time and time series are therefore stopped when the mean is based on fewer than 100 data points. Error bars that indicate twice the standard error of the mean are given at day 30 and 60 for both the LB (red) and the full domain (green). These are distinguished by using offsets of +2 days for red and -2 days for green. Error bars are only included for the 3D particles.*

In Figure 2d, I'm used to seeing wavelet transforms shown with an envelope to indicate the range of validity. Is there an applicable envelope in this case?

Thanks for pointing to this. We interpret the question as being related to a "cone of influence" of the wavelet transform which tells in which range the wavelet transform is influenced by edge effects. Actually, the multivariate ridge analysis automatically performs trimming at the edges of ridges to ensure the masking of such edge effects. In this particular case (Figure 2 in the paper) the drifter was deployed in the Lofoten Basin Eddy, so the drifter was looping right from the start. However, the ridge analysis routine only starts indicating a ridge 2 days after the deployment. This is also about the period the ridge in panel c (black curve) traces at day=0 in the figure (which is actually day 2 after deployment). After the ridge ends around day=63 the drifter times series continues for several hundred days (as does the ridge analysis). The ridge detection at this end is therefore not influenced by edge effects (they're too far away). We will clarify this in the revised manuscript.

There are a number of typos, and I will separately upload a commented version of the pdf, in which I have marked suggested edits.

Thanks. These have been corrected as suggested. An updated manuscript will be accompanied with detailed references to all specific changes.

---

## Author Comment (AC2) · 23 Jan 2021

Author response to reviewer Stefanie Ypma's comments on "The mesoscale eddy field in the Lofoten Basin from high-resolution Lagrangian simulations" by Dugstad et al.

The reviewer's comments are given below in black Times New Roman font, with our response in red Arial font.

Reviewer 2 - Stefanie Ypma

This manuscript by Dugstad et al. presents a thorough analysis of the eddy field in the Lofoten Basin using a multivariate wavelet ridge analysis. Doing so, they've increased the understanding of the formation regions and characteristics of anticyclonic and cyclonic eddies and their respective importance for the heat transport and water mass transformation in the basin. The approach is novel, the paper is very clearly structured and written and regarding the 15 criteria provided for Ocean Science reviewers, I agree with Sarah Gille that the paper is in excellent shape. I would like to add three comments in addition to the issues already raised by Sarah Gille that should be addressed prior to publication.

We thank the reviewer for a detailed reading ad constructive comments on our manuscript. We address each comment as described below.

1. A discussion is missing on the sensitivity of your results to the spatial and temporal seeding distribution of the Lagrangian particles. You seed particles on a 40x40 rectangular grid, but what is the distance between two particles and how does that compare to the average radius of the eddies? In other words, how many particles generally reside in 1 eddy? Regarding the temporal scale, you only seed particles once every week. As you discuss that it is difficult for particles to 'enter' eddies due to high vorticity gradients, aren't you under-sampling the eddy field due to the relatively low seeding frequency?

This is an important issue. The spacing between the drifters is roughly 20 km which is comparable to the average radius of the eddies, meaning that at each seeding round, 4-5 drifters will be deployed in existing eddies (given that the diameter is about 40 km). This will unlikely lead to a problem of under-sampling. Importantly, for any time step a certain fraction of the domain is covered by eddies and a certain fraction is not. By deploying the particles uniformly, we expect the tracers to trace out the area fraction of eddies in a correct manner. This can be seen in the Okubo-Weiss parameter at time=0 in Figure 4 in the manuscript, which shows that the fraction of drifters that are deployed in a region with OW<0 is similar to the fraction of model grid cells that has OW<0 (upper green dashed line compared to green solid line). We therefore conclude

that under-sampling is not an issue, as we have enough seedings and drifters to compare with the model.

The choice of seeding every week is motivated by a hope of achieving a certain level of statistical independence. Note that we seed every week for three years, leading to 156 weeks of seeding. The longer lifetime of an eddy, the more drifters would trace it on its way, but an eddy with a lifetime of about a month would be captured by 20-25 drifters plus the ones that might enter after deployment. Given that there are many eddies that are tracked by roughly 60.000-70.000 drifters for each deployment depth (from Table 2 in the manuscript), the statistics should be well covered when we compute different characteristics of the eddies.

We thus argue that our seeding strategy can be defended. The fact that particles at early times after seeding appear to be preferentially thrown out of anticyclones and drawn into cyclones (compare blue and red dashed lines in Fig. 4) points to interesting real dynamics of which we merely speculate about here. But a typical early-time drop of particles residing in anticyclones from about 0.22 to about 0.14-15 can not explain e.g. the very small relative small contribution from anticyclones (or cyclones) to fluxes shown in Fig. 13.

But your concern of possible sampling issues is one we take very seriously. A revised manuscript will contain more detailed information about the drifter seeding in the Data and Methods section. The revised text will also raise and discuss these issues even more overtly in the Conclusion section.

2. You mention that you don't add any diffusivity, so the particle displacement is purely advective. As the Lofoten Basin is characterised by strong heat losses, there is quite some convection going on. How well can your particles describe vertical motions and temperature changes of water parcels if this convective behaviour is not included?
This is a good point, and we agree that there are issues with reproducing the 'true' vertical motion of water parcels. But first: Our ROMS model has a very high spatial resolution and also employs the GLS vertical mixing scheme. So we are fairly confident that the resolved velocity field (and hydrographic field, hence baroclinic currents) of the model is of high quality, including vertical motions associated with the mesoscale eddy field. A check of the lateral spreading, against real drifter observations was done by Dugstad et al. (2019). It is nonetheless true that our drifter trajectories lack the impact of unresolved vertical motion, including some sub-mesoscale flows and all of small scale turbulent mixing. To add vertical diffusion as a random walk would be an option. But tuning of such a random walk parametrization for realistic simulations is a

fairly involved endeavour which is undergoing active debate within the Lagrangian community.

We have here resorted to the intuitive expectation that the integrated effects of the unresolved vertical motion would primarily lead to a *larger spread* of vertical motion. The kinematic boundary condition at the sea surface is the one obvious place where the (unresolved) mixing might cause a gradual deepening—for all drifter categories (AF, AC and C). In the updated manuscript we will add comments on this issue in the Data and Methods section, in the specific discussion around vertical movement (related to e.g. Fig. 12) and also in the Discussion section.

3. Some of the figures can be improved by adding more clear labels. Comments on the figures, and some other minor comments are marked in the supplement.
Thanks for the input. We will update the labels on the axis and colorbars as you suggest. However, we believe there are some changes that will not lead to a more clear figure. One example of this is Figure 12 in the paper where you suggest we should add legends for both 2D and 3D drifters. Since 2D and 3D have the same color, we find it sufficient to only include 3D drifters in the legend and instead mention in the caption that the 2D results are dashed. This makes the legend smaller such that we can find a good fit for it in the figure.

Some comments to your suggestions/comments in the supplement:

Line 219: Would this not change if you deploy particles at the same time resolution as you have your model-output? So every 6 hour instead of every week?
No, we believe the OW-fraction would be the same independent of deployment frequency. This would only lead to more drifters, but the fraction (the relative amount of drifters in eddies compared to not in eddies) would be the same.

Line 247: How do these results (differences between AC and C's) compare to other studies that estimate eddy characteristics from observations e.g. Sandalyuk et al. (2020)? Could these results be dominated by a larger number of particles that may be residing in the Lofoten Vortex? (something that is also discussed in the Sandalyuk paper). Also, are these differences between the two eddy types something that is specific for the Lofoten Basin, or is this similar in other areas?
In agreement with Raj et al. (2016) we estimate that AC's have longer lifetimes than C's. Furthermore, our estimates indicating larger radius of anticyclones compared to cyclones are in agreement with Raj et al. (2016) and Volkov et al. (2015). We also estimate more anticyclones than cyclones in agreement with

Volkov et al. (2015) and Sandalyuk et al. (2020). You are right that the more elongated shape of cyclones compared to anticyclones can partly be due to many drifters that reside in the Lofoten Basin Eddy (which is circular and anticyclonic). However, Figure 7,e,f in the manuscript suggests that the ellipse linearity is larger for cyclones compared to anticyclones also outside the central Lofoten Basin. The elongated shape of cyclones compared to anticyclones therefore seem to be a more ubiquitous feature. In the revised text we will expand the comparison with other studies of cyclone-anticyclone asymmetries.

All of your other minor comments have been followed up on. An updated manuscript will be accompanied with detailed references to all specific changes.

Paper:
*Dugstad et al. (2019): Vertical Structure and Seasonal Variability of the Inflow to the Lofoten Basin Inferred From High-Resolution Lagrangian Simulations*

---

## Author Response (AR2)

**Author response to the reviewers' comments on "The mesoscale eddy field in the Lofoten Basin from high-resolution Lagrangian simulations" by Dugstad et al.**

We thank both reviewers for constructive comments that helped to improve the manuscript. Below we give a point-by-point response to all questions and comments from the reviewers. Reviewers' comments are reproduced in black (Times New Roman font) with our response following in red (Arial font). When we refer to line numbers, figures or sections, we implicitly refer to the revised manuscript unless anything else is specified. Detailed changes can also be found in the marked-up version of the revised manuscript. We note that the last comments from both reviewers (dated 17th of March 2021) also have been considered and taken care of.

We wish to inform both reviewers that some major changes have been done. First, after comments by Sarah Gille, we now follow the identity of the particles when we compute temperature and vorticity fluxes in an updated Figure 13. Because of this, the earlier categories (AM-AM, R-R, AC-AC, C-C, AC-C and C-AC) are no longer used (see further response below). Furthermore, the fairly confusing Monte Carlo approach was abandoned and the fluxes are now instead computed from three years separately and shown as a mean of these three years. Standard errors of the mean are shown as errorbars. Therefore, while no main conclusions are altered, both the explanation of the method as well as the interpretation/discussion of the results have undergone quite some big changes (see Section 3.5). As a consequence of this, some minor changes in Section 4.1, 4.2 and 4.3 has been changed accordingly.

We also want to mention that the discussion about filaments, and in particular the discussion about the threshold value of |f/2| has been toned down. The threshold was a fairly arbitrary choice and not supported by the litterature. So a proper investigation of this is therefore deferred to later work. Now, we limit the discussion to the observation that the large vorticity fluxes from the ambient flow category likely implies small scales, e.g. filaments, rather than large-scale mean flow. Finally, working through this revision convinced us that our primary focus should be on 3D drifters, as they are likely the best representation of what water parcels experience. So the discussion on 2D drifters has also been toned down somewhat.

Reviewer 1 – Sarah Gille

This manuscript uses a novel approach to assess the contributions of eddies to the Lofoten Basin. In this study, Lagrangian particles are simulated numerically using the

ROMS model and are diagnosed using multivariate wavelet ridge analysis, an approach which allows the authors to readily identify the presence of coherent vortex like structures. The manuscript was written as part of the lead author's PhD thesis, which I also had the pleasure of reviewing, and in a second reading, I remain impressed by the effectiveness of the analysis approach and the clear delineation of contributions from anticyclonic vortices, cyclonic vortices, and ambient flow. The approach is effective, the results are clearly presented, and the findings will be relevant to readers of OceanScience.

I have considered the 15 criteria provided for Ocean Science reviewers, and on thewhole, I think the manuscript is in excellent shape.

We thank you for your constructive comments. They have led to quite a few changes, including a completely new flux calculation and a new Figure 13. Specific comments are addressed below.

There are a few issues that should be addressed prior to publication.

First, the analysis in section 3.5 examines the net transport into and out of the Lofoten Basin due to ambient flow, anticyclonic vortices, and cyclonic vortices. Although the analysis approach is clever and original, I think that it runs the risk of overinterpreting effects. The analysis pairs separate trajectories for flow into the basin and flow out of the basin to consider the net impact on the basin. However, as the authors note, very few particles actually transition from being in an anticyclonic flow on entrance to a cyclonic flow on exit (or vice versa). If particles don't actually experience this change, then using a bootstrapping approach to assess the net contribution due to this unrealistic scenariio seems risky. The manuscript would be stronger if the authors simply examined the net flux into the domain from each of the categories of particles and then separately examined the net flux out of the domain from each category of particle. (Alternatively, if pairing particles at entrance with particles at exit seems imperative, then this should be done using single particles only, without randomly matching entrance particles with other exit particles.).

Your comment here is certainly warranted, and we have thought quite a bit about how to present and interpret these kinds of estimates. But we do believe it is important to present temperature and vorticity fluxes in a near mass-conserving framework. We have therefore avoided to compute fluxes from the entries and exits separately. However, your point that very few drifters actually switch from e.g. anticyclones to cyclones is one we've taken note of. In a revised calculation we have therefore decided to follow your second suggestion, that is, to follow the identity of each drifter and to compute net fluxes (flux in – flux out) for each of these.

Specifically, we now compute the temperature/vorticity fluxes from drifters for the following 6 categories:
- Drifters that enter in anticyclones, and exit as any of the three categories (ACi)
- Drifters that enter in cyclones, and exit as any category (Ci)
- Drifters that enter with the ambient flow, and exit as any category (AFi)
- Drifters that enter as any category, but exit in anticyclones (ACo)
- Drifters that enter as any category, but exit in cyclones (Co)
- Drifters that enter as any category, but exit with the ambient flow (AFo)

The results are given in Figure 1 below - this is the new Figure 13 in the manuscript (note that we will now use 'AF' for ambient flow, and not 'AM', due to comments from reviewer Stefanie Ypma).

The results from this alternate calculation are largely in agreement with the previous calculation in that both heat and vorticity fluxes into the central Lofoten Basin seem to be dominated by the ambient flow. Note that we will now show and discuss flux estimates only for 3D drifters as we believe these are the better representation of real flow paths. For Figure 12 of the manuscript (see below) we compare 3D and 2D drifters to investigate the relation between vertical movement and temperature change. But we cannot think of similar reason for keeping the 2D drifters in the discussion of fluxes. This leads to our also omitting reference to the 2D estimates when we later discuss the contribution of filaments.

[Figure]

*Figure 1: Temperature (a,c,e) and vorticity (b,d,f) fluxes for 3D drifters that are deployed at (a,b) 15 m, (c,d) 200 m and (e,f) 500 m. The number of observations are given with red triangles and are the same for both the temperature and vorticity fluxes. Thick black edges on the AFi and AFo categories in panel (b) and (d) indicate that the bars are given as 1/3 of their actual size. Error bars indicate twice the standard error of the mean. Abbreviations are: AFi=Ambient flow in, ACi=Anticyclones in, Ci=Cyclones in, AFo=Ambient flow out, ACo=Anticyclones out, Co= Cyclones out.*

About the explanation of Figure 13 (Figure 1 in this response letter) we have inserted the following in L399-L419 in the manuscript:

"The procedure is to identify the identity of all drifters that passed through the LB (so both entering and exiting) and to compute the net temperature and vorticity flux into the basin as the difference between fluxes in and fluxes out for each drifter. Note that a drifter can enter and exit the LB several times and we thereby compute fluxes for all drifter segments in the basin. We interpret each drifter as carrying a given mass, and by doing the calculation only on drifters that entered and then exited the basin the calculation approximately conserves mass. For each entry/exit we obtain the values of temperature T and velocity u= (u,v) and estimate the temperature flux into or out of the basin TF=uT·n, where n is the local normal vector to the basin contour (pointing inwards so that entries are positive). For each drifter segment the net temperature flux is then computed as

$TF_{in} - TF_{out}$. The drifters can enter and exit as ambient flow, anticyclones or cyclones. However, it is possible that a drifter changes its category in the basin, meaning that it may for instance enter while being trapped within an anticyclone but exit as part of the ambient flow. We are interested in whether such transitions may play a role in the dynamics and therefore separate the calculations of $TF_{in} - TF_{out}$ into 6 categories: ambient flow in, anything out (AFi); anticyclones in, anything out (ACi); cyclones in, anything out (Ci); anything in, ambient flow out (AFo); anything in, anticyclones out (ACo); anything in, cyclones out (Co). The categories are defined such that the total number of drifter segments in the AFi, ACi and Ci categories equals the total number of drifter segments in the AFo, ACo and Co categories. Furthermore, since the drifters are deployed in 1996, 1997 and 1998 with roughly the same number of drifters deployed each year, we compute net temperature and vorticity fluxes for drifters deployed in each of these years. From these three years we thereby compute a mean temperature and vorticity flux together with standard errors. The results are shown in Figure 13 with error bars given as twice the standard error to indicate 95% confidence intervals. Note that the total fluxes summed over the AFi, ACi and Ci categories equal the fluxes summed over the AFo, ACo and Co categories for each year. However, for the 3-year mean there will be small differences since the number of drifters interacting with the LB is only approximately the same each year."

Related to this, at about line 390, the authors explain the use of a bootstrapping routine to estimate the contributions of particles of different types to the net flux. It's not clear that a bootstrapping approach is necessarily needed for this. If the statistics are relatively Gaussian, then it should be sufficient to compute the mean temperature flux and the standard error of the mean, without needing to go through the computational effort to compute a large bootstrap sample. If bootstrapping is formally necessary, then a bit more explanation would help readers understand why.

We agree with this comment and also note that choosing 75% of the drifters for each iteration of the bootstrapping was also a fairly random choice. We therefore decided to replace this form of uncertainty estimate with a more classical one. Note that our drifters are deployed every week over three years (1996, 1997 and 1998) with roughly the same amount of drifters deployed each year. Instead of estimating the total temperature and vorticity fluxes through all years, we now split the drifters into three groups based on which year they are deployed. We thereby get three independent estimates for the net temperature/vorticity fluxes (from drifters deployed in 1996, 1997 and 1998). In our new Figure 13 we show the 3-sample mean and also twice the standard errors (as a 95% confidence interval for the mean). Through this procedure, the relative importance between ambient flow and eddies is still clear. We describe

how the error bars are computed in L413-L419 in the manuscript (also given here in the response letter above).

Figures 5 and 9 are identified as probability density functions, but neither appears to be normalized so that area under the curve integrates to one. Either they should not be labeled as pdfs (perhaps "distributions of relative frequency"?) or the plotted curves should be normalized by bin width, so that integral of the area under the curve is one.
Agreed. We decided to keep the curves but to refer to these as "relative frequency distributions" (with "PDFs" changed to "RFDs", accordingly).

Figure 12 shows line plots that would be enhanced if statistical uncertainties could be added to the lines. This wold allow readers to judge when the LB region differs statistically from the full domain.
Thanks for the input. We have now computed the standard error and included these as vertical error bars. To be consistent with Figure 13, the error bars here also show twice of the standard error (to indicate a 95% confidence interval for the mean). We first tried to plot the standard error as a shadow in the background, but had to abandon this approach due to a large number of curves that made the figure crowded. Therefore, we have computed and plotted the error bars for specific days (30 and 60) after the drifters entered the basin. To distinguish results from the basin (red) and the full domain (green), error bars are plotted with an offset of +2 days for red and -2 days for green. For better visibility we show error bars only for 3D results in an attempt to keep the figures from getting too busy. The new figure will look like the following:

[Figure]

*Figure 2: Time series of (a,b,c) mean vertical displacement for 3D drifters, (d,e,f) mean temperature change  and (g,h,i) mean density change for 2D (dashed) and 3D (solid lines) drifters. Analyses for the LB (red) and the full domain (green) are shown for the (a,d,g) AF drifters, (b,e,h) cyclonic ridges (C) and (c,f,i) anticyclonic ridges (AC), and for different deployment depths (15m, 200 m and 500 m). To distinguish the drifters that were deployed at 15 m, 200 m and 500 m, an offset of -15 m, -200 m and -500 m is used for the vertical displacements, 0, -2°C and -4°C for the temperature changes and 0, 0.2 kgm⁻³ and 0.4 kgm⁻³ for the density. The mean is based on fewer data points with increasing time and time series are therefore stopped when the mean is based on fewer than 100 data points. Error bars given as twice the standard error of the mean is given at day 30 and 60 for both the LB (red) and the full domain (green). These are distinguished by using offsets of +2 days for red and -2 days for green. Error bars are only included for the 3D particles.*

Note that due to much larger number of AF drifters compared to AC and C, the standard error for the AF category is very small and error bars are therefore hardly visible. The explanation of how the error bars were computed are given in L368-L371:

"Error bars showing twice the standard error of the mean (to indicate 95% significance) plotted at day 30 and 60 after the drifters entered the LB also indicate that the vertical displacements are significant. Note that these are

plotted for cyclonic and anticyclonic ridges, and for the AF drifters, but due to their small magnitudes they are hardly visible for the AF drifters. "

In Figure 2d, I'm used to seeing wavelet transforms shown with an envelope to indicate the range of validity. Is there an applicable envelope in this case?

Thanks for pointing to this. We interpret the question as being related to a "cone of influence" of the wavelet transform which tells in which range the wavelet transform is influenced by edge effects. Actually, the multivariate ridge analysis automatically performs trimming at the edges of ridges to ensure the masking of such edge effects. In this particular case (Figure 2 in the paper) the drifter was deployed in the Lofoten Basin Eddy, so the drifter was looping right from the start. However, the ridge analysis routine only starts indicating a ridge 2 days after the deployment. This is also about the period the ridge in panel c (black curve) traces at day=0 in the figure (which is actually day 2 after deployment). After the ridge ends around day=63 the drifter times series continues for several hundred days (as does the ridge analysis). The ridge detection at this end is therefore not influenced by edge effects (they're too far away). We included the following in L155-157 in the manuscript. "Note here that we have not included a "cone of influence" to indicate the validity range of the wavelet transform. The MWRA routine performs trimming to the ridges, meaning the edges that may be caused by spin-up effects are removed . The ridges are therefore within the valid regime of the wavelet transform."

There are a number of typos, and I will separately upload a commented version of the pdf, in which I have marked suggested edits.

Thanks. These have been corrected as suggested. We also provided information about the spacing of the seeding of particles in the method section. We also note that all references in the text are now listed chronologically by year.

**Reviewer 2 (Stefanie Ypma)**
This manuscript by Dugstad et al. presents a thorough analysis of the eddy field in the Lofoten Basin using a multivariate wavelet ridge analysis. Doing so, they've increased the understanding of the formation regions and characteristics of anticyclonic and cyclonic eddies and their respective importance for the heat transport and water mass transformation in the basin. The approach is novel, the paper is very clearly structured and written and regarding the 15 criteria provided for Ocean Science reviewers, I agree with Sarah Gille that the paper is in excellent shape. I would like to add three comments in addition to the issues already raised by Sarah Gille that should be addressed prior to publication.

Thanks very much for your reading and your comments. We have tried to address each of them throughougly below (and in the revised manuscript).

1. A discussion is missing on the sensitivity of your results to the spatial and temporal seeding distribution of the Lagrangian particles. You seed particles on a 40x40 rectangular grid, but what is the distance between two particles and how does that compare to the average radius of the eddies? In other words, how many particles generally reside in 1 eddy? Regarding the temporal scale, you only seed particles once every week. As you discuss that it is difficult for particles to 'enter' eddies due to high vorticity gradients, aren't you under-sampling the eddy field due to the relatively low seeding frequency?

This is an important issue. The spacing between the drifters is roughly 20 km which is about the average radius of the eddies, meaning that for every seeding 4-5 drifters will be deployed in existing eddies (given that the diameter is about 40 km). However, we do not believe this will lead to a problem of under-sampling. Importantly, for any time step a certain fraction of the domain is covered by eddies and a certain fraction is not. By deploying the particles uniformly, we believe they will trace out the fraction of eddies in a correct manner. This is also shown when investigating the Okubo-Weiss parameter at time=0 in Figure 4 in the manuscript. We see there that the fraction of drifters that are initially deployed in a region with OW<0 is similar to the fraction of grid cells with OW<0 from the model (Upper green dashed line compared to green solid line). We therefore believe under-sampling is not an issue, as we have enough seedings and drifters to compare with the model.

The choice of seeding every week is motivated by a hope of acheiving a certain level of statistical independece. Note that we seed every week for three years, leading to 156 weeks of seeding. The longer lifetime of an eddy, the more drifters would trace it on its way, but an eddy with a lifetime of about a month would thereby be captured by 20-25 drifters plus the ones that might enter after deployment. Given that there are many eddies that are tracked by roughly 60.000-70.000 drifters in total for each level (from Table 2 in the manuscript) we believe the statistics should be well covered when we compute different characteristics of the eddies.

So we believe that our seeding strategy can be defended. The fact that particles at early times after seeding appear to be preferentially thrown out of anticyclones and drawn into cyclones (compare blue and red dashed lines in Fig. 4) points to interesting real dynamics of which we merely speculate about here. But a typical early-time drop of particles residing in anticyclones from about 0.22 to about 0.14-15 can not explain e.g. the relative small contribution from anticyclones (or cyclones) to fluxes shown in Fig. 13.

But your concern of possible sampling issues is one we take very seriously. We have changed Section 2.2 about the Lagrangian simulations to now include the following paragraph in L107-117:

"In all simulations we deploy particles at three levels (15 m, 200 m and 500 m) in sets of 1600 particles every week for three years, from 1 January 1996 to 1 January 1999, with about 20 km spacing between particles (deployment positions are shown in Figure 1 a). In total, this gives 156 weeks of deployments and 1600×156=249,600 particles at each deployment depth. The particles are given a lifetime of 1 year, i.e., the trajectory data end on 1 January 2000. We remove all particles that are deployed in areas shallower than 200 m. After excluding these, the number of particles are reduced to 225,000 at 15 m and 200 m and 195,000 at 500 m for both the 2D and 3D simulation. The spacing and temporal seeding frequency of the particles is designed to achieve relatively uncorrelated motions between different particles, thereby giving independent statistics when we trace and describe the characteristics of eddies and other flow features. With this spacing and temporal seeding frequency, an eddy with an average radius of 20 km and an average lifetime of a month will typically be sampled by 20–25 particles directly deployed in the eddy in addition to the particles that might enter the eddy from outside. For simplicity, we will refer to the Lagrangian particles as "drifters", and use "temperature" and "density" for potential temperature and potential density."

In addition we have inserted in L588-L590 in the Conclusion Section:
"By seeding the drifters uniformly at one-week intervals and with about 20 km spacing, the motion of the drifters could largely be regarded as independent of each other, thereby giving independent statistics of the characteristics of the eddies as well as the ambient flow outside eddies"

2. You mention that you don't add any diffusivity, so the particle displacement is purely advective. As the Lofoten Basin is characterised by strong heat losses, there is quite some convection going on. How well can your particles describe vertical motions and temperature changes of water parcels if this convective behaviour is not included?
This is a good point, and we agree that there are issues with reproducing the 'true' vertical motion of water parcels. But first: Our ROMS model has a very high spatial resolution and also employs the GLS vertical mixing scheme. So we are fairly confident that the resolved velocity field (and hydrographic field, hence baroclinic currents) of the model is of high quality, including vertical motions associated with the mesoscale eddy field. A check on the lateral spreading, against real drifter observations was done by Dugstad et al. (2019). It is nonetheless true that our drifter trajectories lack the impact of unresolved vertical motion, including some sub-mesoscale flows and all of small scale

turbulent mixing. To add vertical diffusion as a random walk would be an option. But tuning of such a random walk parametrization for realistic simulations is a fairly involved endeavour which is undergoing active debate within the Lagrangian community. We have here resorted to the intuitive expectation that the integrated effects of the unresolved vertical mixing would primarily lead to a *larger spread* of vertical motion. The kinematic boundary condition at the sea surface is the one obvious place where the (unresolved) mixing might cause a gradual deepening—for all drifter categories (AF, AC and C).

We have changed the first paragraph in Section 2.2, and in L99-106 we now write:
"We do not add explicit lateral or vertical diffusion to the drifters. The ocean model is very high resolution, and a comparison between synthetic 2D trajectories and real 2D surface drifters have shown that lateral relative dispersion is well reproduced (Dugstad et al., 2019b). It could be questioned whether not adding vertical diffusion can lead to a misleading representation of the vertical motion of the particles. However, to tune such diffusion (implemented as a random walk) is a fairly complex endeavour and is often omitted for such high-resolution modelling (Gelderloos et al., 2017; Dugstad et al., 2019b; Wagner et al., 2019). We essentially believe that adding vertical diffusion would lead to a larger spread of the particles in the vertical but not significantly affect the systematic behavior of the vertical motion of the flow. We return to this issue in the conclusion section."

In the conclusion section we now also include a paragraph discussing the caveats of our approach. There we have inserted in L 619-627:
"Another caveat regarding our simulations is the fact that we did not include the effects of unresolved vertical mixing to the particles. The inclusion of such vertical mixing, parametrized as a random walk process, would likely result in a better representation of the net vertical motion experienced by water parcels. However, calibration of such a parametrization in high-resolution models that already resolve the mesoscale and also part of submesoscale motions is far from trivial. We have resorted to the intuitive expectation that adding parametrized vertical diffusion would likely cause a larger vertical spread of the particles, possibly also a net deepening of particles deployed at 15 m due to the kinematic boundary condition at the surface. But we also believe that most of the systematic results found here, e.g. a stronger deepening of particles in the LB compared to the surroundings as well as stronger deepening in anticyclones compared to the cyclones and the ambient flow, are robust features of the dynamics resolved by our very high-resolution ocean model."

3. Some of the figures can be improved by adding more clear labels. Comments on the figures, and some other minor comments are marked in the supplement.
Thanks for the input. We have updated the labels on the axis and color bars as you suggested. However, we believe there are some changes that will not lead to a more clear figure. One example of this is Figure 12 in the paper where you suggest we should add legends for both 2D and 3D drifters. Since 2D and 3D have the same color, we find it sufficient to only include 3D drifters in the legend and instead mention in the caption that the 2D results are dashed. This makes the legend smaller such that we can find a good fit for it in the figure. Regarding the velocity field in Figure 8 we believe it is most important to have the same scale for the arrows in all panels. We considered to change the style of the arrowheads, but decided to abandon this approach as the arrow heads of small arrows (in panel b,c,d) were still small. We have therefore played around with the scale of the arrows to make small arrow heads more visible together with keeping large arrows (in panel a) small enough to not blur the picture. We decided to increase the scale slightly.

Some comments to your suggestions/comments in the supplement: Here, line numbers refer to the supplement from you.

Line 219: Would this not change if you deploy particles at the same time resolution as you have your model-output? So every 6 hour instead of every week?
No, we believe the OW-fraction would be the same independent of deployment frequency. This would only lead to more drifters, but the fraction (the relative amount of drifters in eddies compared to not in eddies) would be the same. We have included comments about the seeding frequency and how we believe our deployment strategy will make the motion of the particles uncorrelated of each other, leading to independent statistics, in both the method (L112-116) and conclusion section (L 588-590, also given above in the response of your first point in this response letter).

Line 247: How do these results (differences between AC and C's) compare to other studies that estimate eddy characteristics from observations e.g. Sandalyuk et al. (2020)? Could these results be dominated by a larger number of particles that may be residing in the Lofoten Vortex? (something that is also discussed in the Sandalyuk paper). Also, are these differences between the two eddy types something that is specific for the Lofoten Basin, or is this similar in other areas?
In agreement with Raj et al. (2016) we estimate that AC's have longer lifetimes than C's. Furthermore, our estimates indicating larger radius of anticyclones compared to cyclones are in agreement with Raj et al. (2016) and Volkov et al. (2015). We also estimate more anticyclones than cyclones in agreement with Volkov et al. (2015) and Sandalyuk et al. (2020). You are right that the more

elongated shape of cyclones compared to anticyclones can partly be due to many drifters that reside in the Lofoten Basin Eddy (which is circular and anticyclonic). However, Figure 7,e,f in the manuscript suggests that the ellipse linearity is larger for cyclones compared to anticyclones also outside the Lofoten Basin. The elongated shape of cyclones compared to anticyclones is therefore a result of processes also occurring outside the Lofoten Basin. In the old manuscript, it was mentioned in L247 that cyclones had a more elongated shape than anticyclones when discussing Figure 5. Furthermore, when investigating the spatial distribution of the ellipse linearity in Figure 7, we wrote in L276-278 that "over the slope (1000 and 2000 m isobath) off the Lofoten Escarpment, the cyclonic ridges show smaller radii and more elongated shape (higher λ), resulting in a more unstable character......" Given the aim of our study, we believe this provide enough information about the question whether the different characteristics of anticyclones and cyclones are specific for the Lofoten Basin or if they also occur in other regions in the domain.

L260: This provides some great insight into formation regions! Could you add which percentage of particles have a ridge directly after deployment (so seeded in or close to an eddy)?

This task is unfortunately not trivial. The edges of the ridges are usually impacted by spin-up effects. Through the ridge trimming in the MWRA routine, the edges of the ridges are therefore removed. This leads to the consequence that we will not find ridge points directly after the particles are seeded. This can also be seen if studying the fraction of ridge points in Figure 4c in the manuscript (solid black line). How much of the edges that should be removed is estimated by a certain number of cycles. The fraction of ridge points removed at the edges will be the same for each ridge, but in terms of time, more ridge points would be removed from the edges of long ridges (typically from particles looping around large eddies) compared to short ridges (typically from particles looping around small eddies). We can therefore not state whether the first ridge point indicates if the particles are seeded in an eddy or not. We therefore only discuss possible generation sites of eddies with the added information that we are not sure whether a drifter actually traced the generation of an eddy or if it was deployed in an already existing eddy. To avoid confusion about this we have discussed the "cone of influence" of the wavelet transform in L155-157 (See response to Sarah Gille above). Here we also inserted in L158-L160:

"This is also the reason why the ridge is detected the 9 January although the drifter was deployed in the LBE the 8 January. This is a general feature: Due to the ridge trimming, the MWRA routine will never identify ridges on the day of the deployment for a drifter."

All of your other minor comments have been followed up on. These can be seen in the marked-up version of the revised manuscript.

########### END OF RESPONSE LETTER ########################